# BasePrompt: Self-Prompting Genome Language Models for RNA Fitness Prediction

## Abstract

Genome Language Models (GLMs) pre-trained on trillions of nucleotides already exhibit strong zero-shot RNA fitness predictors, yet they cannot be steered toward a specific assay the way a language model is steered by a prompt. We close this gap by letting GLMs prompt themselves. Our method, BasePrompt, asks GLMs to propose short nucleic-acid prefixes and postfixes that maximally activate the fitness signal for a given sequence. To overcome the causal, forward-only nature of most GLMs, we exploit reverse-complement symmetry and generate upstream as well as downstream prompts without ever updating weights or using labeled variants. For zero-shot RNA fitness prediction on RNAGym, BasePrompt achieves a 6.0% relative improvement over the SOTA Evo2 7B model and 6.6%–16.4% over other GLMs, as measured by Spearman correlation. Auxiliary DNA tasks show the same prompting method compresses native-context information into shorter, model-aligned tokens, boosting pathogenicity classification and next-k-base prediction.

Q: A juggler can juggle 16 balls. Half of the balls are golf balls, and half of the golf balls are blue. How many blue golf balls are there?
**Let's think step by step.**

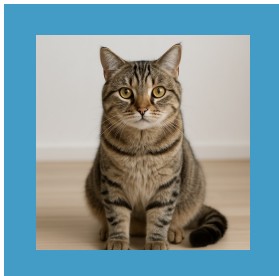
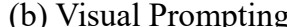
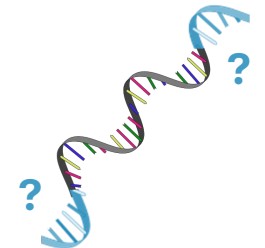

(a) Textual Prompting      (b) Visual Prompting      (c) RNA Prompting

Figure 1: Prompting enhances downstream model performance by guiding within the input space, such as (a) textual prompting (Kojima et al., 2022) and (b) visual prompting (Bahng et al., 2022). However, it is challenging to (c) construct effective prompts for the RNA language.

## 1 Introduction

Despite impressive zero-shot accuracies on dozens of benchmarks (Brixi et al., 2025; Wu et al., 2025; Arora et al., 2025), Genome Language Models (GLMs) still lack a steering mechanism comparable to NLP prompt engineering. In transformers for language or vision a short textual phrase ("Let's think step by step") or a handful of learnt continuous tokens is enough to specialise a model without fine-tuning (Kojima et al., 2022; Brown et al., 2020; Bahng et al., 2022; Jia et al., 2022), as shown in Figure 1. Genome, however, offers no obvious equivalent: human-readable motifs are not necessarily the primitives that GLMs attend to, and domain-specific fine-tuning violates the zero-shot desideratum of RNA-related competitions (Arora et al., 2025). We close this gap by enabling GLMs to write their own prompts. Concretely, we ask them to propose short nucleic-acid prefixes. By "speaking" to themselves in the native genomic language, GLMs expose statistical regularities opaque to humans yet highly predictive, unlocking specialist behavior in a purely zero-shot regime.

In this paper, we propose a novel mechanism, BasePrompt, for generating both upstream and downstream prompts for a given RNA sequence. As illustrated in the left part of Figure 2, our

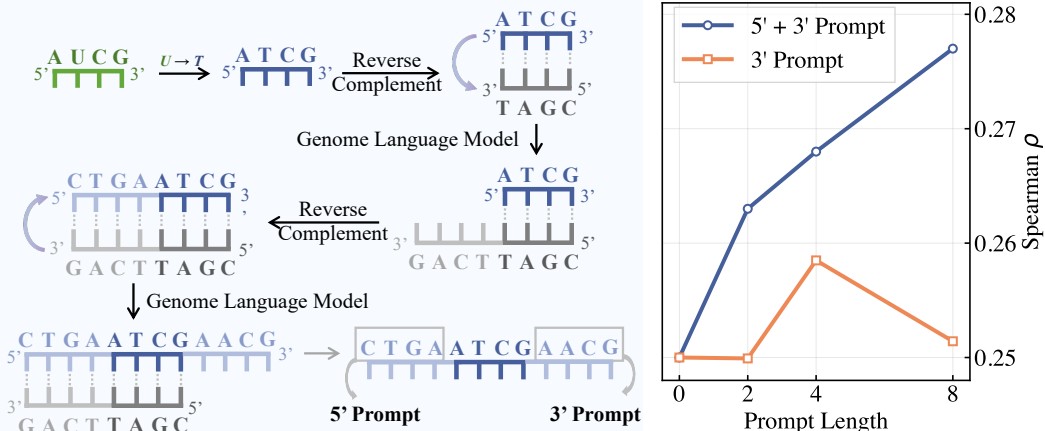

Figure 2: Illustration of BasePrompt. *Left:* It leverages the complementary double-helix structure of DNA to generate upstream ($5'$) and downstream ($3'$) prompts. *Right:* Such a bidirectional prompting has better performance than the pure GLM's downstream prompting in RNA fitness prediction.

approach first converts the RNA sequence to its corresponding DNA sequence. To address the unidirectional nature of causal GLMs, which condition only on upstream context, we use the reverse complement principle of DNA for bidirectional prompting, shown to be more effective in inspiring GLMs in RNA fitness prediction (Arora et al., 2025), as shown in the right panel of Figure 2.

To demonstrate the effectiveness of BasePrompt, we conduct experiments on the RNA fitness prediction benchmark from RNAGym (Arora et al., 2025), which comprises 70 deep mutational scanning assays and over one million RNA sequences. Our method achieves a relative improvement of 6.0 in Spearman's rank correlation compared with previous state-of-the-art results. Furthermore, BasePrompt is highly efficient, as the short prompting is performed only on the 70 reference RNA sequences, not on each of the one million variants. Our method shows consistent gains across multiple GLMs, including Evo2 7B base, Evo2 7B, and Evo2 40B, suggesting robustness to different models.

We conduct additional DNA experiments primarily for analytical purposes. In a zero-shot classification task using the ClinVar dataset (Landrum et al., 2020), our generated short prompts outperform native genomic prompts, particularly under prompt length constraints. It suggests that while native genomic context may contain redundant information, BasePrompt can generate more condensed and model-aligned prompts for predictive utility. For the next $k$-base prediction task (Wu et al., 2025), our upstream prompts guide the model to generate base distributions closer to the native genome.

In summary, our contributions are:

- We introduce BasePrompt, a novel and efficient framework for RNA prompting that leverages GLMs to generate contextual prompts tailored to RNA sequences, significantly improving downstream task performance without requiring additional model training or labeled data.

- Our method establishes a new state-of-the-art in RNA fitness prediction, achieving robust zero-shot improvements on the comprehensive RNAGym benchmark, with consistent performance gains across multiple GLMs in a computationally efficient manner.

- Through auxiliary DNA benchmarks, we demonstrate that prompts generated by BasePrompt are not only highly informationally efficient, but also consistently effective in generative tasks, highlighting their versatility and robustness in biological applications.

## 2 RELATED WORK

### 2.1 DNA REVERSE COMPLEMENT IN MACHINE LEARNING

DNA's inherent double-helical structure (Watson & Crick, 1953) implies that both strands carry equivalent genetic information (Schiff et al., 2024; Marçais et al., 2024; Zhou et al., 2022). However,

standard machine learning architectures do not inherently account for the reverse complement (RC) symmetry (Zhou & Troyanskaya, 2015; Kelley et al., 2016; Quang & Xie, 2019; Li et al., 2023; Clauwaert & Waegeman, 2020). Existing works employed data augmentation (Cao & Zhang, 2019) or embed RC symmetry directly into the model architecture (Shrikumar et al., 2017; Bartoszewicz et al., 2020; Brown & Lunter, 2019; Mallet & Vert, 2021), though such approaches can face optimization difficulties (Zhou et al., 2020). In contrast, our BasePrompt method utilizes the reverse complement principle and GLMs for both the upstream and downstream prompt generation for RNA.

## 2.2 DATA AUGMENTATION FOR DNA

Data augmentation has proven to be a powerful technique in computer vision (CV) and natural language processing to improve model generalization (Alomar et al., 2023; Wang et al., 2024b). Augmentations like flipping in CV (Yang et al., 2022) and synonym substitution in NLP introduce diversity owing to invariance (Bayer et al., 2022). In genomic modeling, augmentation strategies have been tailored to model training, such as reverse complementation (Cao & Zhang, 2019), genomic shifting (de Almeida et al., 2022; Toneyan et al., 2022), random point mutation (Lee et al., 2023), and phylogenetic augmentation (Duncan et al., 2024). In contrast, our BasePrompt adopts the reverse complement and GLMs at test time, requiring no downstream labels, external data, or model tuning.

## 2.3 IN-CONTEXT LEARNING IN BIOLOGY

In-context learning (ICL) is a prominent capability of large language models (Mao et al., 2024; Agarwal et al., 2024). Recently, the ICL paradigm has begun to gain attention in bioinformatics applications (Meier et al., 2021; Liu et al., 2024; Fung et al., 2025). Zero-shot prediction has shown particular promise for predicting the functional impact of genetic variants (Brixi et al., 2025; Benegas et al., 2023; Meier et al., 2021). Some works use crafted natural language instructions and input-output example pairs to steer the model's behavior (Jiang et al., 2025; Wang et al., 2024c). Other works tune the model with motif-oriented pretraining (An et al., 2022) or soft prompting (Nguyen et al., 2023). However, existing ICL methods seem not to work universally with model frozen (Kantroo et al., 2025; Benegas et al., 2025b). In contrast, our BasePrompt leverages the pre-trained GLMs for RNA prompting, bypassing the need for natural language instruction or task examples.

## 3 METHOD

In this section, we describe our RNA prompting method, BasePrompt, which builds upon the causal framework of a genome language model (GLM), as illustrated in Figure 2. In Section 3.1, we discuss the first tackling when given an RNA sequence and the motivation behind it. In Section 3.2, we introduce the causal modeling of DNA generation by GLMs. In Section 3.3, we introduce how our BasePrompt overcomes the inherent direction limitation.

### 3.1 CONVERTING RNA TO DNA

To construct a meaningful prompt for a given RNA sequence, we first map it back to the genome, where contextual information can be naturally derived by Genome Language Models (GLMs). This step is analogous to inverse transcription (Temin & Mizutami, 1970), enabling the RNA to be localized within its genomic coordinates. Since RNA can be directly processed by DNA-based GLMs after a simple substitution of uracil (U) with thymine (T), we hereafter use DNA as the canonical representation for generality. Building on this formulation, we then analyze an autoregressive model's intrinsic reliance on upstream input and explain how BasePrompt generates bidirectional prompts.

### 3.2 AUTOREGRESSIVE MODEL FOR GENOME SEQUENCES

Autoregressive sequence modeling operates on the principle of predicting the next element conditioned on its preceding context. When applied to genome generation, this involves modeling a nucleotide sequence, denoted as $x_1, x_2, \ldots, x_t$, where each token $x_i$ represents a single nucleotide base (*e.g.*, A, T, C, G) or k-mer (*e.g.*, ATGTGG for 6-mer). This modeling is conventionally performed along the $5'$ to $3'$ direction, a choice grounded in fundamental biological processes. The $5'$ and $3'$ ends of a DNA strand, defined by their distinct chemical termini, determine the direction of synthesis. During

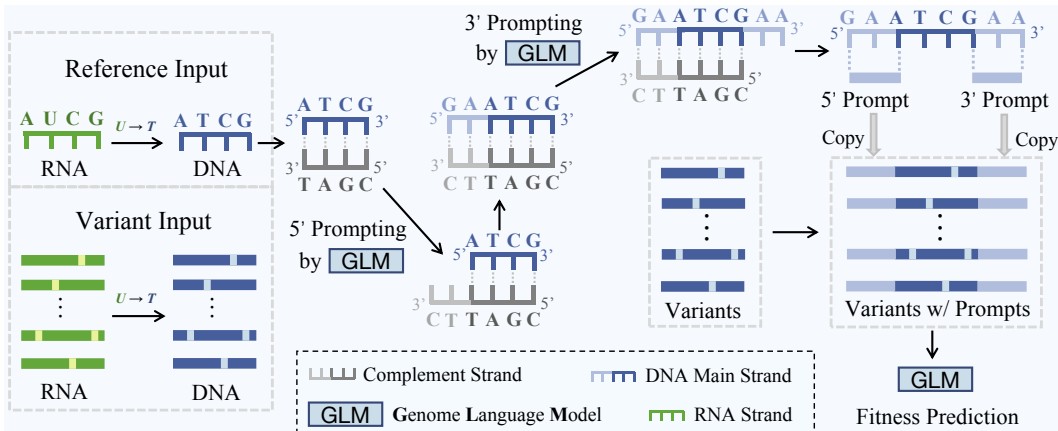

Figure 3: Workflow of BasePrompt for the RNA fitness prediction task. It first converts all input RNA sequences to DNA, replacing U with T. Then, operating solely on the reference DNA sequence, BasePrompt utilizes the complementary strand to perform upstream $5'$ and downstream $3'$ prompting autoregressively by a Genome Language Model (GLM). Generated prompts are then concatenated to the respective $5'$ and $3'$ ends of each variant sequence, creating a full set of prompted variants. Finally, these prompted sequences are fed into the same GLM for fitness prediction.

processes such as DNA replication and transcription, new nucleotides are exclusively added to the $3'$ terminus, thereby extending the chain in a unidirectional $5'$ to $3'$ manner.

For autoregressive generation, the model predicts the next nucleotide token $x_{t+1}$ conditioned on the sequence $x_1, x_2, \ldots, x_t$, formally expressed in Equation (1).

$$P(x_{t+1} \mid x_1, x_2, \ldots, x_t) = \text{softmax}(f(x_1, x_2, \ldots, x_t)) \tag{1}$$

Here, $f(\cdot)$ is a function, typically implemented by a genome language model, that maps the sequence $x_1, x_2, \ldots, x_t$ to logits over the next possible nucleotide token. The softmax function normalizes these outputs into valid probabilities. This process continues iteratively to generate the full sequence.

### 3.3 REVERSE COMPLEMENT AND PROMPTING

We propose that the inherent symmetry of DNA, with its double-stranded structure, may help overcome the prompting direction challenge. In addition to the main strand of DNA, there exists a complementary reverse strand, oriented in the opposite direction ($5'$ to $3'$ opposite to $3'$ to $5'$) of the main strand. This complementary and reverse structure offers an opportunity to extend the context for sequence generation by utilizing the reverse complement strand.

Let $x_1, x_2, \ldots, x_t$ represent the nucleotide sequence of the main strand, from $5'$ to $3'$. The corresponding reverse complement sequence is denoted as $\hat{x}_t, \hat{x}_{t-1}, \ldots, \hat{x}_1$. For example, if $x_i$ is the nucleotide at position $i$ in the main strand, then $\hat{x}_{t-i}$ is the complementary base in the reverse strand at the corresponding position, as shown in Figure 4 (contents in black color).

By transforming to the reverse complement, we enable predictions for the downstream sequence ($3'$) on the reverse complement strand, which corresponds to the upstream ($5'$) sequence on the main strand. Specifically, the prediction on the reverse complement is $P(\hat{x}_{t+1} \mid x_t, x_{t-1}, \ldots, x_1)$. This allows us to predict the *nonexistent* context on the main strand shown in the contents in blue of Figure 4. We assume that the token before $x_1$ is $x_{-1}$, indicating that $x_0$ does not exist.

After generating the downstream of the reverse complement strand, we can map the generated bases to the main strand. Such a newly generated part is actually the upstream prompt of the main strand. Detailed steps are shown in the Appendix C. We formalize such a process in Equation (2).

$$
\begin{array}{ccccccccc}
5'- & \text{C} & \text{T} & \text{A} & \text{T} & \text{G} & & \text{T} & \text{G} & \text{G} & -3' \\
\text{Main Strand} & x_{-2} & x_{-1} & x_1 & x_2 & x_3 & \cdots & x_{t-2} & x_{t-1} & x_t \\
& | & | & | & & | & | & | \\
\text{Reverse Complement Strand} & \hat{x}_{t+2} & \hat{x}_{t+1} & \hat{x}_t & \hat{x}_{t-1} & \hat{x}_{t-2} & \cdots & \hat{x}_3 & \hat{x}_2 & \hat{x}_1 \\
3'- & \text{G} & \text{A} & \text{T} & \text{A} & \text{C} & & \text{A} & \text{C} & \text{C} & -5'
\end{array}
$$

Generated by GLM

Figure 4: DNA double strands described in sequence generation.

$$
\begin{aligned}
P(x_{-1} \mid x_t, x_{t-1}, \ldots, x_1) &= \mathrm{compl}(P(\hat{x}_{t+1} \mid x_t, x_{t-1}, \ldots, x_1) \\
&= \mathrm{compl}(\mathrm{softmax}(f(\hat{x}_1, \hat{x}_2, \ldots, \hat{x}_t)))
\end{aligned}
\tag{2}
$$

The $\mathrm{compl}(\cdot)$ represents the probability of the complementary bases, and the function $f(\cdot)$ here is the same as in Equation (1), as genome language models are pre-trained on both strands (Brixi et al., 2025; Schiff et al., 2024; Wu et al., 2025). By incorporating the reverse complement strand, we can effectively generate bidirectional prompts. Let $N$ denote the number of tokens of the prompt, and we can now formalize generation as shown in Equation (3).

$$
P(x_{t+1} \mid x_{-N}, x_{-(N-1)}, \ldots, x_{-1}, x_1, x_2, \ldots, x_t)
\tag{3}
$$

According to the introduction above, we can now leverage the DNA symmetry to generate upstream prompts during inference by sampling on the next-nucleotide probability. Notably, this symmetry is bidirectional, meaning we can start from the reverse complement strand and, by reversing the process, operate again on the main strand. More details can be found in Appendix C.

## 4 EXPERIMENT

To evaluate the effectiveness of BasePrompt, we conduct experiments in a zero-shot setting and utilize only the model inference without any tuning.

### 4.1 EXPERIMENTAL SETTINGS

We introduce the tasks, datasets, models, and baselines as follows. The detailed information of our four datasets is depicted in Table 1. Further experimental details are provided in Appendix B.

**Task-A (main): RNA fitness prediction.**  We evaluate BasePrompt by predicting the functional effect of RNA sequences in a zero-shot manner on the RNAGym benchmark (Arora et al., 2025). It comprises 70 DMS assays, covering over one million variants across diverse RNA types and species, with sequence lengths ranging from 45 to 5,592 bases. We employ four metrics: Spearman's rank correlation (Spearman), Area Under the Receiver Operating Characteristic Curve (AUC), Matthews Correlation Coefficient (MCC), and Area Under the Precision-Recall Curve (AUPRC).

**Task-B (auxiliary): DNA fitness prediction.**  We take the ClinVar dataset (Landrum et al., 2020) to compare BasePrompt with the native prompt from the genome. The task is predicting the pathogenicity of human clinical single-nucleotide variants. For ClinVar, we adopt the version from Benegas et al. (2025a)[1], which contains 40,976 samples. Our metrics include Spearman and AUPRC.

**Task-C (auxiliary): DNA next-base prediction.**  Since DNA has the "ground truth" prompt which RNA does not have, we further evaluate the synthetic prompt of BasePrompt on the DNA next-base prediction task (Wu et al., 2025)[2], which is then filtered to retain sequences containing only A, G, C, and T. The task requires the model to predict the next $N$ bases given an input DNA sequence. We evaluate generation quality using three metrics that capture different aspects of performance. (1) Accuracy: The percentage of correctly predicted bases, calculated as $(N_{correct}/N) \times 100\%$. (2) GC

---

[1]https://huggingface.co/datasets/songlab/clinvar
[2]https://huggingface.co/datasets/GenerTeam/next-kmer-prediction

Table 1: Information about our datasets. RNAGym serves as the main dataset for evaluation, while ClinVar and the next $k$-base prediction (NBP) are used as auxiliary datasets for analysis.

| Dataset | Modality | Task | Species | Size |
|---------|----------|------|---------|------|
| RNAGym | RNA | Fitness Prediction | Eukaryote, Prokaryote, Virus | 1,117,995 |
| ClinVar | DNA | Fitness Prediction | Human | 40,976 |
| NBP | DNA | Generation | Eukaryote, Prokaryote, Virus | 84,648 |

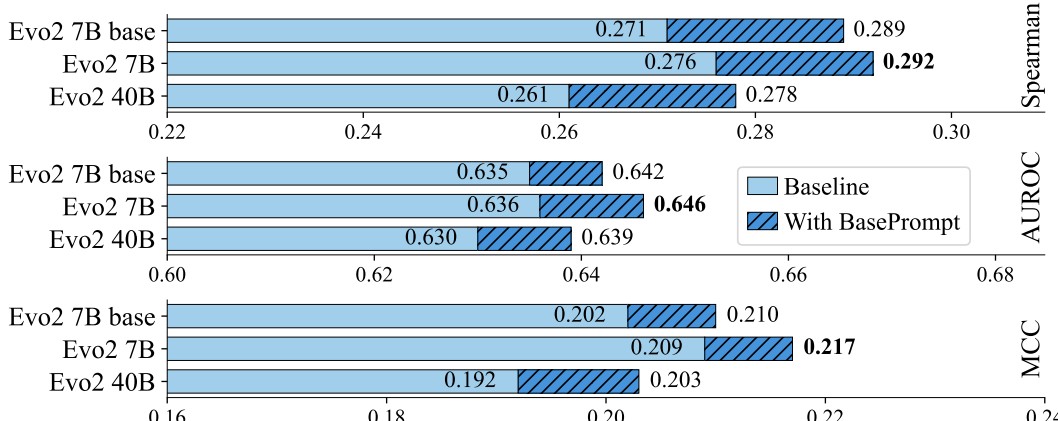

Figure 5: BasePrompt improves performance across diverse models on the RNAGym benchmark. We evaluate our method on the Evo families with the same prompt length of three. Bold numbers highlight the best performance.

Content Difference: The absolute difference between the GC ratio of the generated bases and the ground-truth bases averaged on all samples. (3) K-mer Jensen-Shannon Divergence (JSD): The JSD between the k-mer frequency distributions of the generated and ground-truth sequences.

**Model and sampling.** We conduct experiments using recently released genome language models (GLMs) to ensure a comprehensive evaluation across different architectures and scales. Specifically, we employ five models in RNA fitness prediction: Evo1 (Nguyen et al., 2024), Evo1.5 (Merchant et al., 2024), and Evo2 (Brixi et al., 2025) (7B base, 7B, 40B). For Evo models, which use single-nucleotide tokenization, we restrict generation to the four valid DNA bases and set top-$k$=1, *i.e.*, greedy strategy. In DNA tasks, we also include the GENERator (Wu et al., 2025) models (1.2B, 3B), which use 6-mer tokenization. We set sampling hyperparameters as temperature=1.0, top-$k$=4, and top-$p$=1.0. The same GLM is used for both prompting and performing the downstream task.

## 4.2 BASEPROMPT IMPROVES RNA FITNESS PREDICTION

For RNA fitness prediction, Genome Language Models (GLMs) compute the negative log-likelihood of the entire sequence $S$, and the length-normalized average serves as the predicted fitness score $f(S)$ (Brixi et al., 2025). An ensemble score, $(f(S) + f(S_{rc}))/2$, can be obtained by averaging with the reverse complement, $S_{rc}$. In the official RNAGym benchmark, different models were inconsistently reported—Evo1.5 with direct prediction and Evo2-7B with RC ensemble. Since we observed that RC ensembling has a non-trivial effect, we re-implemented all baselines and reported ensemble results for fair comparison. We report the ensembled results in the main text, and the direct predictions are provided in Appendix D.

**BasePrompt achieves state-of-the-art performance on RNAGym.** Our method brings consistent improvement across three metrics on the Evo series, as illustrated in Figure 5. We utilize a prompt length of three for these models. The performance of additional baseline models is sourced from the RNAGym paper (Arora et al., 2025), including Nucleotide Transformer (NT) (Dalla-Torre et al.,

Table 2: BasePrompt improves performance across diverse models and achieves SOTA results on the RNAGym benchmark. Our method achieves relative improvements of 6.0–16.4% in Spearman $\rho$, 1.2–3.9% in AUROC, and 4.0–24.3% in MCC over the respective baseline models. We use the reverse complement ensemble for both baselines and ours. Gray rows indicate our results, while bold numbers denote the best performance. Results of each model's best prompt length are reported.

| Model Name | Spearman $\rho$ ↑ | AUROC ↑ | MCC ↑ |
|---|---|---|---|
| RNA-FM | 0.217 | 0.605 | 0.155 |
| RNAErnie | 0.189 | 0.598 | 0.154 |
| NT | 0.163 | 0.577 | 0.115 |
| RiNALMo | 0.155 | 0.582 | 0.122 |
| GenSLM | 0.122 | 0.558 | 0.083 |
| Evo1 | 0.192 | 0.590 | 0.131 |
| w/ BasePrompt | 0.211 | 0.604 | 0.146 |
| Evo1.5 | 0.199 | 0.594 | 0.147 |
| w/ BasePrompt | 0.232 | 0.617 | 0.183 |
| Evo2 7B base | 0.271 | 0.635 | 0.202 |
| w/ BasePrompt | 0.289 | 0.643 | **0.219** |
| Evo2 7B | 0.276 | 0.636 | 0.209 |
| w/ BasePrompt | **0.292** | **0.646** | 0.217 |
| Evo2 40B | 0.261 | 0.630 | 0.192 |
| w/ BasePrompt | 0.288 | 0.645 | 0.212 |

Table 3: BasePrompt consistently outperforms the baseline in RNAGym across diverse sampling hyperparameters, achieving relative improvements ranging from 1.6% to 7.2% in Spearman's rank correlation. All experiments are conducted using the Evo2 7B with a prompt length of three.

| Top-$k$ | Top-$p$ | Spearman $\rho$ | AUROC | MCC |
|---|---|---|---|---|
| Baseline | | 0.276 | 0.636 | 0.209 |
| 1 | - | 0.292 | 0.646 | 0.217 |
| 2 | - | 0.296 | 0.648 | 0.223 |
| 3 | - | 0.295 | 0.648 | 0.221 |
| 4 | - | 0.280 | 0.641 | 0.207 |
| - | 0.3 | 0.282 | 0.641 | 0.211 |
| - | 0.5 | 0.296 | 0.648 | 0.222 |
| - | 0.7 | 0.282 | 0.641 | 0.211 |
| - | 0.9 | 0.272 | 0.637 | 0.209 |

2025), RiNALMo (Penić et al., 2025), RNA-FM (Chen et al., 2022), RNAernie (Wang et al., 2024a), and GenSLM (Zvyagin et al., 2023). Among these, BasePrompt on Evo2 7B achieves the highest performance, with a relative improvement of 6% in Spearman's rank correlation.

**BasePrompt performs even better with more flexible prompt lengths.** In previous experiments, we reported the result of a fixed length of three for consistency, as shown in Figure 5. However, BasePrompt can be more powerful when equipped with a selective length for each model. As illustrated in Table 2, our method achieves relative improvements of 6.0–16.4% in Spearman $\rho$, 1.2–3.9% in AUROC, and 4.0–24.3% in MCC over the respective baseline models. It demonstrates the generality and potential of our method. The prompt lengths of BasePrompt for models on the table from top to bottom are 32, 32, 5, 3, and 5.

**BasePrompt is robust to sampling hyper-parameters of GLM generation.** Our method consistently outperforms the baseline across various top-$k$ and top-$p$ values, as shown in Table 3. We use

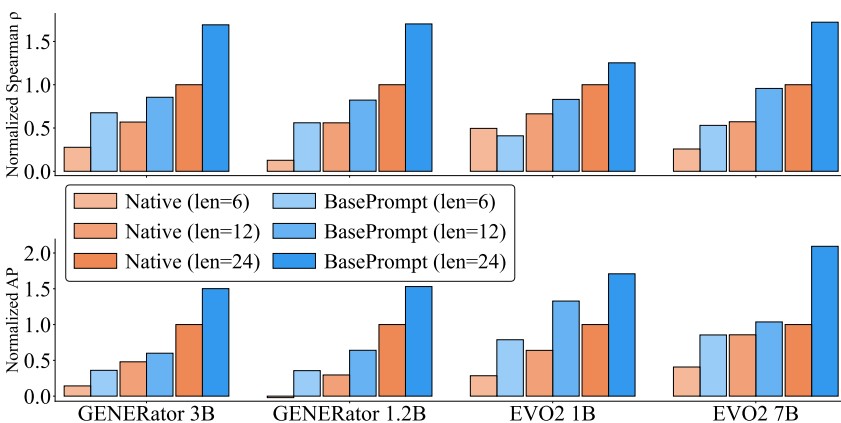

Figure 6: BasePrompt outperforms the native genome prompt on the ClinVar dataset under limited expansion. The "len" denotes the prompt length. For fair comparison, we normalize Spearman's rank correlation (Spearman $\rho$) and average precision (AP) by setting the value at input length 510 (prompt length=0) to 0.0, and the value at prompt length of 24 to 1.0.

greedy sampling (top-$k = 1$) in the main experiment for controlled comparison without randomness and get a Spearman's $\rho$ of 0.287 on Evo2 7B. However, the top-$k = 2$ brings a better Spearman's $\rho$ of 0.297. It demonstrates the hidden potential of our BasePrompt.

**BasePrompt is computationally efficient for large-scale fitness prediction.** We only generate a sequence with three bases 70 times, a negligible computational overhead when compared to the 1,117,995 model inferences required for fitness prediction across all variants. Additionally, the extra 6 bases (3+3) introduced by our prompt contribute only 1.5% to the overall computational cost of fitness prediction, which is derived by averaging the number and length of sequences for each assay.

### 4.3 AUXILIARY EXPERIMENTS ON DNA FOR ANALYSIS

To further analyze BasePrompt, we turn to DNA-based tasks where the genome could provide natural prompts. We concentrate on the performance of discriminative and generative tasks, but not interpretable motifs, since the prompt is used for steering downstream GLMs but not for humans.

#### 4.3.1 BASEPROMPT AIDS CLINICAL VARIANT EFFECT PREDICTION

On the ClinVar dataset (Landrum et al., 2020), the model is given a variant-containing sequence and a matched reference, and must classify the variant as pathogenic or benign in a zero-shot setting. Following Evo2's setup (Brixi et al., 2025)[3], the variant effect is measured by the log-likelihood difference between the mutant and reference. For comparison, we consider three inputs: (i) the original sequence $S_{\text{origin}}$, (ii) native prompt with real genomic prompt, $S_{5'\text{native}} + S_{\text{origin}} + S_{3'\text{native}}$, and (iii) prompt generated by our BasePrompt, $S_{5'\text{our}} + S_{\text{origin}} + S_{3'\text{our}}$.

**BasePrompt outperforms native genomic prompt under short prompts.** As shown in Figure 6, orange bars represent native prompt ($S_{5'\text{native}} + S_{\text{origin}} + S_{3'\text{native}}$) and blue bars represent BasePrompt ($S_{5'\text{our}} + S_{\text{origin}} + S_{3'\text{our}}$). For comparability, we normalize Spearman's rank correlation and average precision by setting the score at no prompting($S_{\text{origin}}$) to 0.0 and the score at a prompt length of 24 per side ($S_{5'\text{native}} + S_{\text{origin}} + S_{3'\text{native}}$) to 1.0. Notably, our generated prompts consistently outperform native prompts when the prompt length is less than or equal to 24 bases on each side.

We hypothesize that GLMs, through large-scale pretraining with next-token prediction, acquire the ability to reconstruct "efficient" contextual signals. In contrast, a native prompt may contain redundant information. Since the same GLM serves as both prompt generator and predictor, the synthesized prompt may have bridged the distributional gap closer to the model's pretraining distribution.

---

[3]The public source code in Github can be found here.

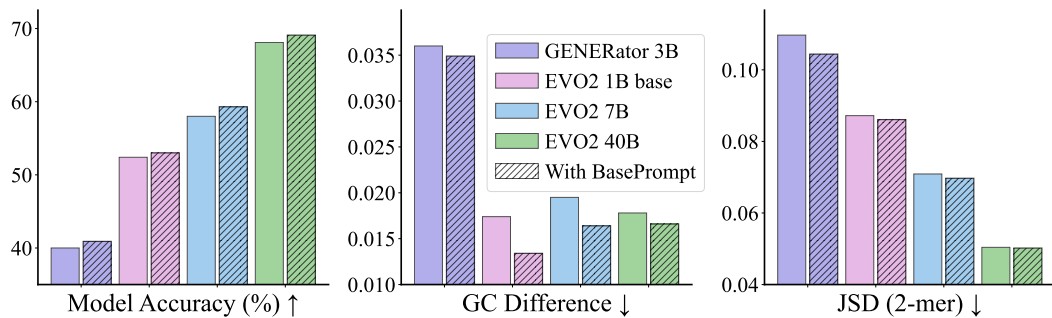

Figure 7: The bases predicted by BasePrompt show a closer alignment with the genome. Bold numbers represent our performance, while the regular numbers correspond to the raw input results. BasePrompt yields a higher accuracy, lower GC difference, and 2-mer Jensen-Shannon Divergence.

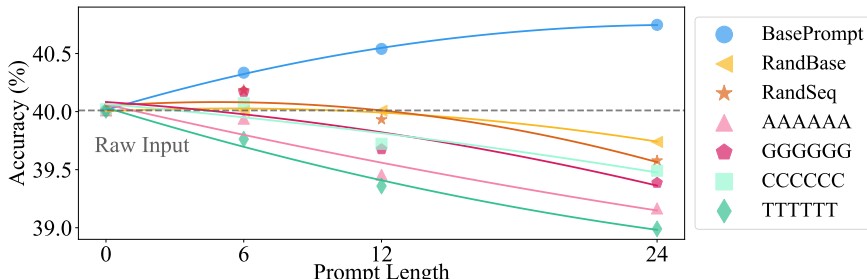

Figure 8: BasePrompt indicates generating a positively meaningful prompt, rather than just random expansion. The task is to predict the next 6 bases given a sequence of length 192.

### 4.3.2 ASSESSING THE BIOLOGICAL PLAUSIBILITY OF GENERATED PROMPTS

It is also necessary to evaluate whether the generated sequences remain biologically plausible. We adopt the dataset from Wu et al. (2025) and then restricts it to canonical nucleotides (A, T, C, G). We ask GLMs to predict the next $k$ bases prediction, conditioned on a 192-base input sequence.

**BasePrompt yields generations closely aligned with biologically plausible patterns.** As shown in Figure 7, our method consistently achieves higher base-level accuracy and smaller GC-content difference compared to direct prediction on raw input, even under the $5'$ prompt of 192 bases beyond the original input. Furthermore, the 2-mer JSD remains lower across species groups, indicating that the generated prompt better preserves local sequence statistics. Additional results for varying prompt lengths, input lengths, GLMs, and sampling strategies are provided in Appendix E.

**BasePrompt generates more informative prompts compared to other prompting strategies.** We compare the performance of BasePrompt with various prompting approaches in Figure 8. The RandomSeq is to randomly generate a sequence and prepend it to the original input. We modify this prompting by selecting the bases from the original sequence, *i.e.*, the RandomSeq strategy. Additionally, we explore fixed-sequence prompts, such as repeating A/G/C/T.

## 5 CONCLUSION

We present BasePrompt, a novel framework that leverages Genome Language Models (GLMs) to generate bidirectional RNA prompts, enhancing performance in downstream tasks without requiring domain-specific fine-tuning. By enabling GLMs to write their own prompts, we close a critical gap in the application of prompting to RNA sequences. Our method achieves state-of-the-art results in RNA fitness prediction and demonstrates robustness across multiple GLM models. Additionally, we show that the generated prompts are efficient and effective, offering consistent improvements in various genomic tasks. BasePrompt represents a significant step toward unlocking the full potential of GLMs.

## ETHICS STATEMENT

This research adheres to the ICLR Code of Ethics. No human subjects were involved, and all datasets used are publicly available. The study focuses on computational methods for RNA sequence prompting and does not raise privacy, security, or ethical concerns related to personal data. There are no conflicts of interest or sponsorships affecting this work, and all experiments comply with legal and ethical standards.

## REPRODUCIBILITY STATEMENT

To ensure the reproducibility of our work, we provide the source code and all relevant materials in the supplementary materials. A detailed README file is included to guide users through the process of reproducing our results step by step. The code and data processing steps for the RNA fitness prediction experiments are available, along with instructions for setting up the environment and running the experiments. We also include descriptions of the datasets used and any assumptions made during the study to ensure full transparency.

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

# APPENDIX

In the appendix, we first discuss the use of Large Language Models (LLMs) in Appendix A. We then detail the experimental settings in Appendix B. Next, we provide an in-depth description of the BasePrompt process in Appendix C. Additionally, we present supplementary experiments related to RNA and DNA in Appendix D and Appendix E, respectively.

## A    THE USE OF LARGE LANGUAGE MODELS

In the process of writing this paper, large language models (LLMs) played a significant role in supporting various tasks. Specifically, GPT-5 (OpenAI, 2025) was used to assist with grammar checking and error correction, refining wording and phrasing, and providing suggestions for visualizations of tables and figures. These tools were employed to enhance the clarity, coherence, and presentation of the content. The LLMs were primarily used for tasks that involved improving the linguistic quality of the text and offering guidance on the effective visualization of data, but they did not contribute to the scientific ideation or the formulation of research hypotheses. The contributions of these models are acknowledged to ensure transparency in the research process.

## B    EXPERIMENTAL SETTINGS

All experiments are conducted in a zero-shot setting, utilizing only the inference capabilities of pre-trained models without any fine-tuning. We design a comprehensive suite of four tasks to evaluate BasePrompt across both generative and discriminative capabilities on DNA and RNA sequences. The tasks are detailed in Appendices B.1 to B.3. We then describe the models and hyperparameters in Appendix B.4 and elaborate on the core zero-shot mechanism for variant effect prediction in Appendix B.5.

### B.1    TASK-A: RNA VARIANT EFFECT PREDICTION ON RNAGYM

**Task and Dataset**    We evaluate BasePrompt on the RNA variant effect prediction (VEP) task using the RNAGym benchmark (Arora et al., 2025). The objective is to predict the functional effect of RNA sequence variants in a zero-shot manner. Performance is evaluated by comparing the predicted functional ranking of variants against experimental measurements from deep mutational scanning (DMS) assays. The benchmark comprises 70 standardized DMS assays, covering 1,117,995 variants across diverse RNA types (*e.g.*, mRNA-splicing, tRNA, aptamer, ribozyme) and species (eukaryote, prokaryote, virus, human), with sequence lengths ranging from 45 to 5,592 bases.

**Evaluation and Metrics**    We employ four complementary metrics. Spearman's rank correlation assesses the monotonic relationship between predicted scores and experimental measurements. Area Under the Receiver Operating Characteristic Curve (AUC) evaluates the model's ability to discriminate between functional and non-functional variants. Matthews Correlation Coefficient (MCC) provides a balanced measure of classification quality, even for imbalanced datasets. Area Under the Precision-Recall Curve (AUPRC) is particularly informative for imbalanced data, focusing on the performance of identifying positive instances. To ensure a balanced evaluation across RNA types with varying numbers of assays, we first compute the average performance for each RNA type and then report the mean of these per-type averages.

### B.2    TASK-B: NEXT-BASE PREDICTION

**Task and Dataset**    We assess the generative capabilities of models on a DNA next-base prediction task, using data from various species, including eukaryote, prokaryote, virus, and human (Wu et al., 2025; Sanabria et al., 2024). The task requires the model to predict the next $N$ bases given an input DNA sequence. We use the dataset from (Wu et al., 2025)[4], filtered to retain only sequences containing A, G, C, and T, resulting in 84,648 sequences. Dataset statistics are provided in Table 5.

---

[4]https://huggingface.co/datasets/GenerTeam/next-kmer-prediction

Table 4: We provide details on deep mutational scanning assays in RNAGym (Arora et al., 2025), complementing the information on each RNA assay that is not included in the original paper.

| RNA Type | Dataset | Sequence Number | Sequence Length |
|---|---|---|---|
| Aptamer (Target binding ability) | Tome_2014_GFP_aptamer | 417 | 82 |
| | Tome_2014_NELFE_aptamer | 2652 | 70 |
| Ribozyme (Cleavage and splicing) | Andreasson_2020_ribozyme | 7343 | 74 |
| | Beck_2022_ribozyme | 21321 | 69 |
| | Janzen_2022_fam1a1_ribozyme | 1953 | 71 |
| | Janzen_2022_fam1b1_ribozyme | 1953 | 71 |
| | Janzen_2022_fam21_ribozyme | 1953 | 71 |
| | Janzen_2022_fam22_ribozyme | 1953 | 71 |
| | Janzen_2022_fam31_ribozyme | 1953 | 71 |
| | Kobori_2015_ribozyme_j12 | 255 | 230 |
| | Kobori_2015_ribozyme_p4 | 255 | 190 |
| | Kobori_2015_ribozyme_tw | 1023 | 77 |
| | Kobori_2016_osa_ribozyme | 10296 | 54 |
| | Kobori_2018_ribozyme | 16383 | 160 |
| | McRae_2024_5tu_ribozyme | 74942 | 152 |
| | McRae_2024_t1_ribozyme | 47503 | 135 |
| | Peri_2022_ribozyme | 16383 | 197 |
| | Pitt_2010_ribozyme | 186 | 87 |
| | Roberts_2023_HDV_ribozyme | 33930 | 87 |
| | Roberts_2023_cepeb3_ribozyme | 21321 | 69 |
| | Roberts_2023_hh_ribozyme | 9045 | 45 |
| | Roberts_2023_hp_ribozyme | 22578 | 71 |
| | Roberts_2023_tw_ribozyme | 10296 | 48 |
| | Soo_2021_ribozyme | 64019 | 425 |
| | Zhang_2020_cpeb3_ribozyme | 111417 | 81 |
| | Zhang_2024_OR4K15_ribozyme | 61393 | 140 |
| | Zhang_2024_line1_full_ribozyme | 69583 | 146 |
| | Zhang_2024_line1_mini_ribozyme | 149710 | 46 |
| mRNA Coding (Coding mRNA fitness) | A0A2Z5U3Z0_9INFA_Doud_2016 | 10715 | 1698 |
| | BCHB_CHLTE_Tsuboyama_2023_2KRU | 1652 | 156 |
| | BLAT_ECOLX_Firnberg_2014 | 4783 | 861 |
| | BLAT_ECOLX_Jacquier_2013 | 989 | 861 |
| | BRCA1_HUMAN_Findlay_2018 | 886 | 5592 |
| | C6KNH7_9INFA_Lee_2018 | 10754 | 1701 |
| | CALM1_HUMAN_Weile_2017 | 1813 | 450 |
| | CAPSD_AAV2S_Sinai_2021 | 42328 | 2208 |
| | CBS_HUMAN_Sun_2020 | 7217 | 1656 |
| | CCDB_ECOLI_Adkar_2012 | 1176 | 306 |
| | DLG4_RAT_McLaughlin_2012 | 1576 | 2175 |
| | DOCK1_MOUSE_Tsuboyama_2023_2M0Y | 3099 | 198 |
| | ESTA_BACSU_Nutschel_2020 | 2172 | 639 |
| | F7YBW8_MESOW_Ding_2023 | 7922 | 282 |
| | GFP_AEQVI_Sarkisyan_2016 | 51714 | 717 |
| | HECD1_HUMAN_Tsuboyama_2023_3DKM | 6835 | 216 |
| | IF1_ECOLI_Kelsic_2016 | 1367 | 219 |
| | MLAC_ECOLI_MacRae_2023 | 4007 | 636 |
| | MTHR_HUMAN_Weile_2021 | 12464 | 1971 |
| | OBSCN_HUMAN_Tsuboyama_2023_1V1C | 3421 | 195 |
| | OXDA_RHOTO_Vanella_2023_activity | 6396 | 1092 |
| | P53_HUMAN_Kotler_2018 | 1048 | 1182 |
| | PKN1_HUMAN_Tsuboyama_2023_1URF | 1305 | 213 |
| | POLG_DEN26_Suphatrakul_2023 | 16897 | 2700 |
| | POLG_PESV_Tsuboyama_2023_2MXD | 6287 | 159 |
| | PSAE_PICP2_Tsuboyama_2023_1PSE | 1616 | 204 |
| | PTEN_HUMAN_Mighell_2018 | 7260 | 1212 |
| | Q837P4_ENTFA_Meier_2023 | 697 | 1770 |
| | R1AB_SARS2_Flynn_2022 | 5725 | 918 |
| | RCRO_LAMBD_Tsuboyama_2023_1ORC | 2401 | 189 |
| | RDRP_I33A0_Li_2023 | 12003 | 2271 |
| | RNC_ECOLI_Weeks_2023 | 4277 | 681 |
| | SPIKE_SARS2_Starr_2020_binding | 3802 | 3822 |
| | SPTN1_CHICK_Tsuboyama_2023_1TUD | 3485 | 180 |
| | SR43C_ARATH_Tsuboyama_2023_2N88 | 1663 | 144 |
| | SUMO1_HUMAN_Weile_2017 | 1700 | 306 |
| | TAT_HV1BR_Fernandes_2016 | 1577 | 258 |
| mRNA Splicing (Splicing ability) | Julien_2016_mRNA | 189 | 63 |
| | Ke_2017_mRNA | 5559 | 51 |
| tRNA (Stability and growth) | Domingo_2018_tRNA | 4175 | 72 |
| | Guy_2014_tRNA | 25491 | 105 |
| | Li_2016_tRNA | 65536 | 72 |

**Evaluation and Metrics** We evaluate generation quality using three metrics that capture different aspects of performance. (1) Accuracy: The percentage of correctly predicted bases, calculated as $(N_{correct}/N) \times 100\%$. (2) GC Content Difference: The absolute difference between the GC content

Table 5: Information of the Next-base Prediction dataset.

| Subset | Taxomony / Species Type | Size |
|---|---|---|
| Eukaryote | Archaea, Viruses, Bacteria, Eukaryota | 29,944 |
| Bacteria | Bacteria | 30,801 |
| Others | Mitochondrion, Archaea, Plasmid, Virus, Plastid | 23,903 |

ratio of the generated bases and the ground-truth bases. (3) k-mer Jensen-Shannon Divergence (JSD): The JSD between the k-mer frequency distributions of the generated and ground-truth sequences. These metrics provide a multi-grained analysis: accuracy measures base-level precision, GC content difference reflects compositional distribution, and k-mer JSD captures local sequence statistics.

### B.3 TASK-C: CLINICAL VARIANT EFFECT PREDICTION

We further evaluate BasePrompt on a discriminative task focused on predicting the pathogenicity of human clinical single-nucleotide variants (SNVs). We use the ClinVar dataset (Landrum et al., 2020) to assess generalization on clinically significant variations. The model is given a pair of sequences—one containing the SNV and a corresponding reference—and must classify the variant as pathogenic or benign in a zero-shot setting. We adopt the version from (Benegas et al., 2025a)[5], which contains 40,976 samples.

**Evaluation and Metrics**  Our evaluation metrics include Spearman and AUPRC. Spearman's Rank Correlation quantifies the monotonic relationship between the model's predicted pathogenicity scores (*e.g.*, logits or probabilities) and the ground-truth binary labels. It assesses how well the model preserves the inherent ranking of variants by their severity, indicating if higher scores generally correspond to more severe variants. AUPRC is particularly informative for imbalanced datasets, such as clinical variant annotations, where pathogenic variants are often rarer. It focuses on the trade-off between precision and recall for the positive class (pathogenic variants), providing a more sensitive measure of performance when identifying these critical cases.

### B.4 MODELS AND HYPERPARAMETERS

We conduct experiments using recently released genome language models (GLMs) to ensure a comprehensive evaluation across different architectures and scales. Specifically, we employ six models from two families: GENERator-1.2B, GENERator-3B (Wu et al., 2025), Evo2-1B-base, Evo2-7B, Evo2-7B-base, and Evo2-40B (Brixi et al., 2025). For generative tasks, we employ standard sampling strategies: temperature, top-k, and top-p. Our default settings are chosen based on the models' tokenization schemes. For GENERator models, which use 6-mer tokenization, we set temperature=1.0, top-$k$=4, and top-$p$=1.0. For Evo2 models, which use single-nucleotide tokenization, we restrict generation to the four valid DNA bases (A, T, C, G) by setting top-k=1, effectively performing greedy decoding. The robustness of BasePrompt to variations in sampling strategies is further discussed and evaluated in Table 11.

BasePrompt is a pure inference-time approach that enhances model performance by generating the prompt, without altering the underlying model architecture or its downstream task pipeline. The specific application of BasePrompt to each task is detailed in Appendix C. For consistency and fair comparison, the same GLM is used for both generating the bidirectional prompts and for performing the final downstream task. For instance, if an experiment is labeled Evo2 7B, it signifies that Evo2 7B was leveraged for both prompting and the subsequent prediction.

### B.5 ZERO-SHOT MECHANISM OF VARIANT EFFECT PREDICTION

In this section, we introduce the detailed mechanism of zero-shot implementation on variant classification. In a zero-shot setting, the model is not explicitly trained on labeled variant effect data but instead leverages its pretrained knowledge of genomic sequences to assess mutation impact directly

---
[5]Dataset link.

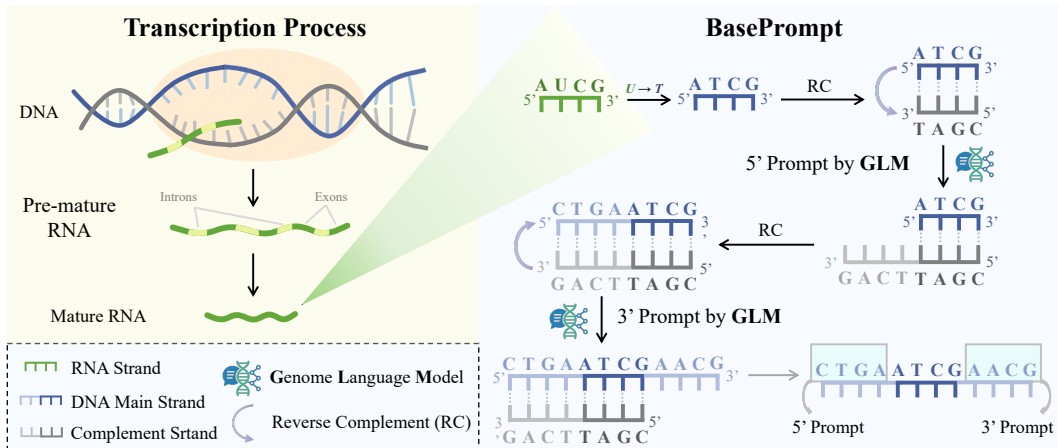

Figure 9: Motivation and illustration of BasePrompt. The transcription from DNA to RNA involves mechanisms such as alternative splicing, which can irreversibly limit the available sequence context. Inspired by the complementary double-helix structure of DNA, we introduce BasePrompt, a novel method for RNA prompting. It generates the bidirectional prompts by autoregressively predicting the upstream $5'$ and downstream $3'$ prompts directly on the main strand. Our method is inference-time and model-agnostic, making it both convenient and effective.

Figure 10: Input DNA sequence described in Markov chain.

| | $5'-$ | A | T | G | | T | G | G | $-3'$ |
|---|---|---|---|---|---|---|---|---|---|
| *Main Stand* | | $x_1$ | $x_2$ | $x_3$ | $\cdots$ | $x_{t-2}$ | $x_{t-1}$ | $x_t$ | |

from sequence likelihoods. The input to the model consists of a pair of sequences: a reference sequence (ref) and a variant sequence (var), differing by a single nucleotide variant (SNV).

Here, the reference sequence $\mathbf{x}^{\text{ref}}$ represents the wild-type (non-mutated) version of a genomic region, while $\mathbf{x}^{\text{var}}$ is an otherwise identical sequence that contains an SNV at a specific position. The task is to predict whether the variant results in loss of function (LoF) or the opposite (Non-LoF). Given a reference sequence $\mathbf{x}^{\text{ref}}$ and a variant sequence $\mathbf{x}^{\text{var}}$, we compute a log-likelihood score for each sequence by averaging the model's log-probabilities over all positions:

$$\log p(\mathbf{x}) = \frac{1}{L} \sum_{t=1}^{L} \log p(x_t \mid \mathbf{x}_{<t}) \tag{4}$$

Here, $L$ is the sequence length, and $p(x_t \mid \mathbf{x}_{<t})$ is the probability assigned to the true nucleotide at position $t$ under the model's autoregressive output. In practice, this is computed by applying a log-softmax over the model's output logits at each position and gathering the value corresponding to the ground-truth token. Then the delta likelihood score between the reference and variant is calculated as:

$$\Delta\mathcal{L} = \log p(\mathbf{x}^{\text{ref}}) - \log p(\mathbf{x}^{\text{var}}) \tag{5}$$

This score serves as a proxy for mutation impact, with higher values indicating greater disruption under the model's learned distribution. To evaluate the classification performance of this approach, the Area Under the Receiver Operating Characteristic (AUROC) is computed based on the delta likelihood score.

Figure 11: DNA double strands described in a Markov chain.

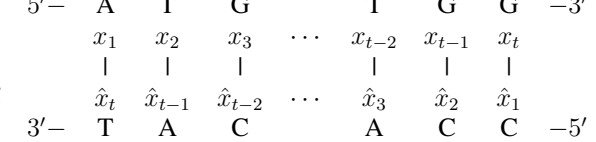

Figure 12: Generating the downstream of the reverse complement strand.

## C BasePrompt Mechanism

We describe the detailed process involved in generating the DNA strands based on the Markov chain model outlined in the main text. The overview is illustrated in Figure 9. This process involves inputting an initial DNA sequence, generating the reverse complement strand, and mapping generated bases to the main strand using the principle of complementary base pairing. The methodology involves simulating both the main and reverse complement strands, as shown in Figure 10, Figure 11, Figure 12, and Figure 13.

For detailed application in generation and variant classification tasks, we describe the usage of BasePrompt in Figure 3. We define the expansion of the original $5' - sequence \rightarrow 3'$ by GLM as $3'$ prompting, and the reverse complement of $5' \leftarrow sequence - 3'$ by GLM as $5'$ prompting, details of which are shown in Figure 3. The upper section illustrates the $5'$ prompting for the generative task. Additionally, further inference can facilitate the formulation of a bidirectional prompting. The lower section depicts the variant classification process, where both the $5' + 3'$ prompts are applied to the reference sequence. These prompts are then concatenated to the respective $5'$ and $3'$ ends of each variant sequence. For computational efficiency, we perform bidirectional prompting only on the reference sequence. These prompts are then applied consistently across all associated variant sequences.

**Step 1: Input DNA Sequence.**   The input DNA sequence is represented as a series of states in a Markov chain model. The main strand of the DNA sequence is defined in Figure 10. This sequence corresponds to the states $x_1, x_2, x_3, \ldots, x_t$ of the Markov chain. The model processes the sequence by transitioning from one state to the next.

**Step 2: DNA Double-Strand Representation.**   Using the initial sequence, the model generates both the main and reverse complement (RC) strand. The RC strand is derived by replacing each base in the main strand with its complement, shown in Figure 11. The corresponding Markov chain states are given by $x_1, x_2, \ldots, x_t$ for the main strand and $\hat{x}_1, \hat{x}_2, \ldots, \hat{x}_t$ for the RC strand.

**Step 3: Generating Downstream of the Reverse Complement Strand.**   To generate the downstream sequence of the reverse complement strand, the model applies transitions based on previous transition probabilities. The downstream sequence is generated as shown in Figure 12.

**Step 4: Mapping the Generated Reverse Complement Bases to the Main Strand.**   Finally, the model maps the generated bases of the reverse complement strand back to the main strand. This is done using mapping rules derived from the transitions in the Markov model shown in Figure 13.

Figure 13: Mapping the generated bases of the reverse complement to the main strand.

| | $5'-$ | C | T | A | T | G | $\cdots$ | T | G | G | $-3'$ |
|---|---|---|---|---|---|---|---|---|---|---|---|
| *Main Stand* | | $x_{-2}$ | $x_{-1}$ | $x_1$ | $x_2$ | $x_3$ | $\cdots$ | $x_{t-2}$ | $x_{t-1}$ | $x_t$ | |
| | | | | | | | | | | | |
| *Reverse Complement Stand* | | $\hat{x}_{t+2}$ | $\hat{x}_{t+1}$ | $\hat{x}_t$ | $\hat{x}_{t-1}$ | $\hat{x}_{t-2}$ | $\cdots$ | $\hat{x}_3$ | $\hat{x}_2$ | $\hat{x}_1$ | |
| | $3'-$ | G | A | T | A | C | | A | C | C | $-5'$ |

---

**Algorithm 1** Pseudocode of the BasePrompt Algorithm

---

**Input:**
   $S_{rna}$   // The source RNA sequence.
   $N$      // The target prompt length for each side.
   $GLM$    // A pre-trained Genome Language Model (generates from 5' to 3').
   *Helper Functions:*
   **function** RNAtoDNA($S_{in}$): Converts RNA sequence $S_{in}$ to DNA (U → T).
   **function** RevComplement($S_{in}$): Returns the reverse complement of a DNA sequence.
   **function** Generate($S_{in}, N, GLM$): Extends $S_{in}$ and returns **only the newly generated** part.
**Output:** The full sequence $S_{full}$, and the generated prompts $S_{5'}$ and $S_{3'}$

1: **function** BASEPROMPT($S_{rna}, N$)
2:    $S_{main} \leftarrow$ RNAtoDNA($S_{rna}$)                    ▷ Main strand, 5' to 3'

   *// Phase A: 5' Prompt utilizing the complementary strand*
3:    $S_{rc} \leftarrow$ RevComplement($S_{main}$)                ▷ Complement strand, 5' to 3'
4:    $S_{up,rc} \leftarrow$ Generate($S_{rc}, N, GLM$)        ▷ Upstream part on complement strand, 5' to 3'
5:    $S_{5'} \leftarrow$ RevComplement($S_{up,rc}$)          ▷ Final 5' prompt on main strand, 5' to 3'

   *// Phase B: 3' Prompt (autoregressive)*
6:    $S_{prefix} \leftarrow S_{5'} + S_{main}$                    ▷ Main strand, 5' to 3'
7:    $S_{3'} \leftarrow$ Generate($S_{prefix}, N, GLM$)      ▷ Final 3' prompt on main strand, 5' to 3'

   *// Final Assembly*
8:    $S_{full} \leftarrow S_{5'} + S_{main} + S_{3'}$
9:    **return** $S_{full}, S_{5'}, S_{3'}$
10: **end function**

---

# D   SUPPLEMENTED RNA EXPERIMENTS

In this section, we introduce the re-implementation details of RNAGym (Arora et al., 2025) in Appendix D.1, and discuss the direct prediction without ensemble in Appendix D.2.

## D.1   DIFFERENCE OF RE-IMPLEMENTATION AND OFFICIAL RESULTS

As shown in Table 6, different models were inconsistently reported—Evo1.5 with direct prediction and Evo2-7B with reverse complement (RC) ensemble. Since we observed that RC ensembling has a non-trivial effect, we re-implemented all Evo1 (Nguyen et al., 2024), Evo1.5 (Merchant et al., 2024), and Evo2 (Brixi et al., 2025) baselines and reported ensemble results for fairness in the main text.

## D.2   EXPERIMENTS OF DIRECT PREDICTION W/O ENSEMBLE

As shown in Table 7, BasePrompt performs well even using only direct prediction without ensembles. We use the reverse complement ensemble for both baselines and ours on the Evo series, following the Evo2 setting of RNAGym. It demonstrates the generality of our method. The prompt lengths of BasePrompt for models on the table from top to bottom are 32, 32, 3, 2, and 5.

Table 6: Effect of reverse-complement (RC) ensembling on RNAGym. Gray rows highlight the RNAGym-reported results and our corresponding re-implementation under the same setting. RNAGym reports Evo1.5 using direct prediction but Evo2 7B with RC ensemble, introducing inconsistency and bias in the benchmark. Our re-implementation evaluates both settings.

| Model | Setting | Spearman $\rho$ | AUROC |
|---|---|---|---|
| Evo1.5 | RNAGym reported | 0.177 | 0.583 |
| | Our re-impl. (direct) | 0.177 | 0.583 |
| | Our re-impl. (+ RC ensemble) | 0.199 | 0.594 |
| Evo2 7B | RNAGym reported | 0.276 | 0.636 |
| | Our e-impl. (direct) | 0.251 | 0.622 |
| | Our re-impl. (+ RC ensemble) | 0.276 | 0.636 |

Table 7: Our BasePrompt improves RNAGym when using direct prediction without reverse complement ensemble. Specifically, our method achieves relative improvements of 7.2–29.6% in Spearman $\rho$, 1.6–5.5% in AUROC, and 7.8–31.7% in MCC over the respective baseline models. Note that the baseline results of Evo2 (Brixi et al., 2025) are different from RNAGym (Arora et al., 2025) since RNAGym uses both the forward and the reverse chain for function prediction. Results of each model's best prompt length are reported.

| Model Name | Spearman $\rho \uparrow$ | AUROC $\uparrow$ | MCC $\uparrow$ |
|---|---|---|---|
| RNA-FM | 0.217 | 0.605 | 0.155 |
| RNAErnie | 0.189 | 0.598 | 0.154 |
| Nucl. Transformer | 0.163 | 0.577 | 0.115 |
| RiNALMo | 0.155 | 0.582 | 0.122 |
| GenSLM | 0.122 | 0.558 | 0.083 |
| Evo1 | 0.175 | 0.587 | 0.135 |
| w/ BasePrompt | 0.205 | 0.601 | 0.156 |
| Evo1.5 | 0.177 | 0.583 | 0.129 |
| w/ BasePrompt | 0.229 | 0.615 | 0.170 |
| Evo2 7B base | 0.257 | 0.626 | 0.194 |
| w/ BasePrompt | 0.281 | 0.637 | 0.209 |
| Evo2 7B | 0.251 | 0.622 | 0.176 |
| w/ BasePrompt | 0.269 | 0.633 | 0.210 |
| Evo2 40B | 0.264 | 0.630 | 0.195 |
| w/ BasePrompt | **0.286** | **0.641** | **0.216** |

### D.3 PERFORMANCE ACROSS MODELS AT A PROMPT LENGTH OF THREE

As shown in Table 8, we set the prompt length of BasePrompt to three and show that all models have a great performance. Although the result does not demonstrate the best performance for each model, the steady improvement demonstrates the stability of our BasePrompt. Note that we use the reverse complement ensemble for both baselines and ours on the Evo series following RNAGym. Details about direct predictions and reverse complement ensemble can be found in Table 6.

We also compare the performance across RNA types in Table 9 at a prompt length of three. Notably, BasePrompt shows improvements across all RNA types with the Evo2 40B model. In terms of model differences, BasePrompt demonstrates significant gains in mRNA splicing and mRNA coding, highlighting the potential of RNA prompting for mRNA-related functions. For the remaining three

Table 8: BasePrompt improves performance across diverse models and achieves state-of-the-art results on the RNAGym (Arora et al., 2025) benchmark. We use the reverse complement ensemble for both baselines and ours on the Evo series. Gray rows denote our method, while bold numbers highlight the best performance. The prompt length is set to three on both sides.

| Model Name | Spearman $\rho \uparrow$ | AUROC $\uparrow$ | MCC $\uparrow$ |
|---|---|---|---|
| RNA-FM | 0.217 | 0.605 | 0.155 |
| RNAErnie | 0.189 | 0.598 | 0.154 |
| NT | 0.163 | 0.577 | 0.115 |
| RiNALMo | 0.155 | 0.582 | 0.122 |
| GenSLM | 0.122 | 0.558 | 0.083 |
| Evo1 | 0.192 | 0.590 | 0.131 |
| w/ BasePrompt | 0.202 | 0.595 | 0.135 |
| Evo1.5 | 0.199 | 0.594 | 0.147 |
| w/ BasePrompt | 0.205 | 0.597 | 0.147 |
| Evo2 7B base | 0.271 | 0.635 | 0.202 |
| w/ BasePrompt | 0.289 | 0.642 | 0.210 |
| Evo2 7B | 0.276 | 0.636 | 0.209 |
| w/ BasePrompt | **0.292** | **0.646** | **0.217** |
| Evo2 40B | 0.261 | 0.630 | 0.192 |
| w/ BasePrompt | 0.278 | 0.639 | 0.203 |

RNA types, RNA FM outperforms BasePrompt, indicating the importance of incorporating RNA-specific information. A promising future direction could involve exploring ways to integrate RNA-related data into RNA prompting, such as leveraging RNA language models to generate upstream and downstream prompts with BasePrompt.

### D.4 PERFORMANCE AT BEST PROMPT LENGTHS

As illustrated in Table 2, our method achieves relative improvements of 6.0–16.4% in Spearman $\rho$, 1.2–3.9% in AUROC, and 4.0–24.3% in MCC over the respective baseline models. We use the reverse complement ensemble for both baselines and ours on the Evo series, following the Evo2 setting of RNAGym. It demonstrates the generality of our method. The prompt lengths of BasePrompt for models on the table from top to bottom are 32, 32, 5, 3, and 5. We also show the tendency of relative improvement in Figure 14. The height of our increase in purple is much higher across three metrics.

## E SUPPLEMENTED DNA EXPERIMENTS

In this section, we introduce several supplemented DNA experiments. We conduct experiments across a wide range of input sequence lengths for the next-base prediction task[6]. In the following experiments, we will demonstrate the general effectiveness of our BasePrompt across various (1) input lengths, (2) genome language models, and (3) sampling strategies.

**BasePrompt consistently improves accuracy across different prompt lengths.** As illustrated in Figure 15, BasePrompt performs well across a range of input sequence lengths and target sequence lengths. Given the substantial variability in absolute accuracy with respect to input sequence length, we present the accuracy improvement, defined as $\frac{ACC_{BasePrompt} - ACC_{baseline}}{ACC_{baseline}} \times 100\%$, to facilitate a direct comparison across different prompt lengths. The baseline values in Figure 15 correspond to the original input sequence, with $ACC_{baseline}$ values of 32.8%, 35.6%, 40.1%, 45.7%, 52.3%, and 56.6% for increasing prompt length, respectively. Notably, the greatest improvements are observed

---

[6]Dataset version.

Table 9: The average Spearman's rank correlation of BasePrompt scores across various RNA types. Our method demonstrates significant improvements in mRNA splicing and mRNA coding, while RNA FM outperforms in the remaining types. Both our method and the baselines use the reverse complement ensemble in the Evo series, consistent with the Evo2 setting of RNAGym. Gray rows represent our results, with bold numbers highlighting the best performance. The prompt length of each model is three.

| Model Name | mRNA-splic. | tRNA | Aptamer | Ribozyme | mRNA-cod. | Average |
|---|---|---|---|---|---|---|
| RNA FM | 0.103 | **0.464** | **0.190** | **0.201** | 0.129 | 0.217 |
| RNAErnie | 0.230 | 0.416 | 0.037 | 0.142 | 0.117 | 0.189 |
| NT | 0.121 | 0.317 | 0.146 | 0.147 | 0.083 | 0.163 |
| RiNALMo | 0.348 | 0.260 | 0.026 | 0.072 | 0.070 | 0.155 |
| GenSLM | 0.173 | 0.093 | 0.126 | 0.135 | 0.084 | 0.122 |
| Evo1 | 0.142 | 0.422 | 0.067 | 0.150 | 0.177 | 0.192 |
| w/ BasePrompt | 0.188 | 0.409 | 0.080 | 0.153 | 0.179 | 0.202 |
| Evo 1.5 | 0.146 | 0.430 | 0.082 | 0.164 | 0.173 | 0.199 |
| + w/ BasePrompt | 0.179 | 0.424 | 0.085 | 0.164 | 0.173 | 0.205 |
| Evo2 7B Base | 0.387 | 0.420 | 0.079 | 0.186 | 0.284 | 0.271 |
| + w/ BasePrompt | 0.435 | 0.427 | 0.136 | 0.159 | 0.287 | 0.289 |
| Evo2 7B | 0.432 | 0.387 | 0.119 | 0.170 | 0.271 | 0.276 |
| + w/ BasePrompt | **0.468** | 0.413 | 0.140 | 0.167 | 0.274 | **0.292** |
| Evo2 40B | 0.305 | 0.431 | 0.097 | 0.172 | 0.298 | 0.261 |
| + w/ BasePrompt | 0.321 | 0.431 | 0.152 | 0.179 | **0.305** | 0.278 |

Table 10: BasePrompt is generally effective on different models. As the prompt length increases, the performance improvement becomes more pronounced. The task is predicting the next 6 bases given a sequence of length 192. The green value represents the delta compared to the baseline.

| Model | Accuracy (%) by Prompt Length | | | | | |
|---|---|---|---|---|---|---|
| | 0 (Baseline) | 6 | 12 | 24 | 48 | 96 |
| GENERator 1.2B | 38.4 | 38.5 (↑0.1) | 38.4 (↑0.0) | 38.7 (↑0.3) | 39.0 (↑0.6) | 39.4 (↑1.0) |
| GENERator 3B | 40.1 | 40.3 (↑0.2) | 40.5 (↑0.4) | 40.7 (↑0.6) | 40.9 (↑0.8) | 41.6 (↑1.5) |
| Evo2 1B base | 46.0 | 46.3 (↑0.3) | 46.7 (↑0.7) | 47.0 (↑1.0) | 47.1 (↑1.1) | 47.0 (↑1.0) |
| Evo2 7B | 52.8 | 53.1 (↑0.3) | 53.4 (↑0.6) | 54.0 (↑1.2) | 54.2 (↑1.4) | 54.5 (↑1.7) |
| Evo2 40B | 65.4 | 65.7 (↑0.3) | 65.8 (↑0.4) | 66.1 (↑0.7) | 66.4 (↑1.0) | 66.6 (↑1.2) |

for mid-range input sequence lengths. This result aligns with expectations, as shorter input sequences may lack sufficient information to generate accurate prompts during inference.

**BasePrompt demonstrates broad efficacy across diverse model families.** We evaluate Base-Prompt on both the Evo2 and GENERator families, tasked with predicting the next 6 bases from a 192-base sequence, as summarized in Table 10. For both families, sequences utilizing our prompts consistently outperform their corresponding baselines, defined as raw DNA input. Notably, Base-Prompt 's effectiveness is evident across a range of model sizes, from 1.2B to 40B parameters, highlighting its robustness and lack of dependence on a specific model capacity. Furthermore, the improvement over the baseline tends to be more pronounced in larger models, underscoring that BasePrompt can leverage advancements in genome language models.

**BasePrompt is robust across diverse sampling hyperparameters.** We perform a comprehensive evaluation across various sampling hyperparameters, shown in Table 11. Such a sampling process

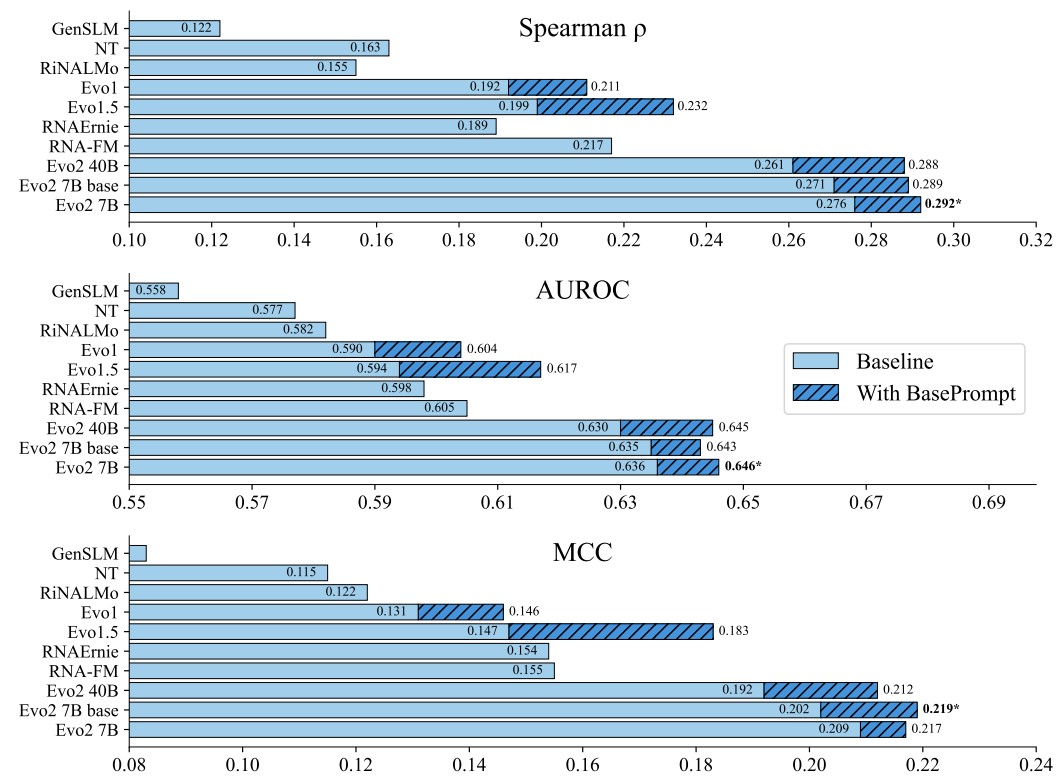

Figure 14: BasePrompt improves performance across diverse models and achieves SOTA results on the RNAGym benchmark. Results of each model's best prompt length are reported. We use the reverse complement ensemble following RNAGym on the Evo series.

Table 11: BasePrompt is robust under diverse generation sampling hyperparameters. The task is predicting the next 6 bases given a DNA sequence of length 384 using GENERator-3B. The top is the default setting in our experiments. The green number is the improvement compared with the baseline.

| Temp | Top-$k$ | Top-$p$ | Accuracy (%) | |
| --- | --- | --- | --- | --- |
| | | | Baseline | BasePrompt |
| 1.0 | 4 | 1.0 | 45.7 | 47.5+1.8 |
| 1.2 | / | 1.00 | 37.4 | 38.9+1.5 |
| 1.0 | / | 1.00 | 39.0 | 40.5+1.5 |
| 0.8 | / | 1.00 | 40.6 | 42.3+1.7 |
| 0.6 | / | 1.00 | 42.6 | 44.2+1.6 |
| 1.0 | 100 | / | 42.7 | 44.1+1.4 |
| 1.0 | 50 | / | 43.4 | 44.9+1.5 |
| 1.0 | 10 | / | 45.1 | 46.2+1.1 |
| 1.0 | 1 | / | 46.4 | 48.2+1.8 |
| 1.0 | / | 1.00 | 39.0 | 40.5+1.5 |
| 1.0 | / | 0.99 | 39.2 | 40.7+1.5 |
| 1.0 | / | 0.90 | 40.4 | 41.5+1.1 |
| 1.0 | / | 0.70 | 41.5 | 43.2+1.7 |

influences both the bidirectional prompting and the next-base prediction task. Our method consistently performs well across a range of temperatures, top-$k$, and top-$p$ values. Here, the symbol "/" denotes a neutral setting, *i.e.*, no restriction in the sampling process (top-$k = 0$ and top-$p = 1$). As top-$k$ and top-$p$ increase, the sampling flexibility increases, followed by a notable drop in next-base prediction

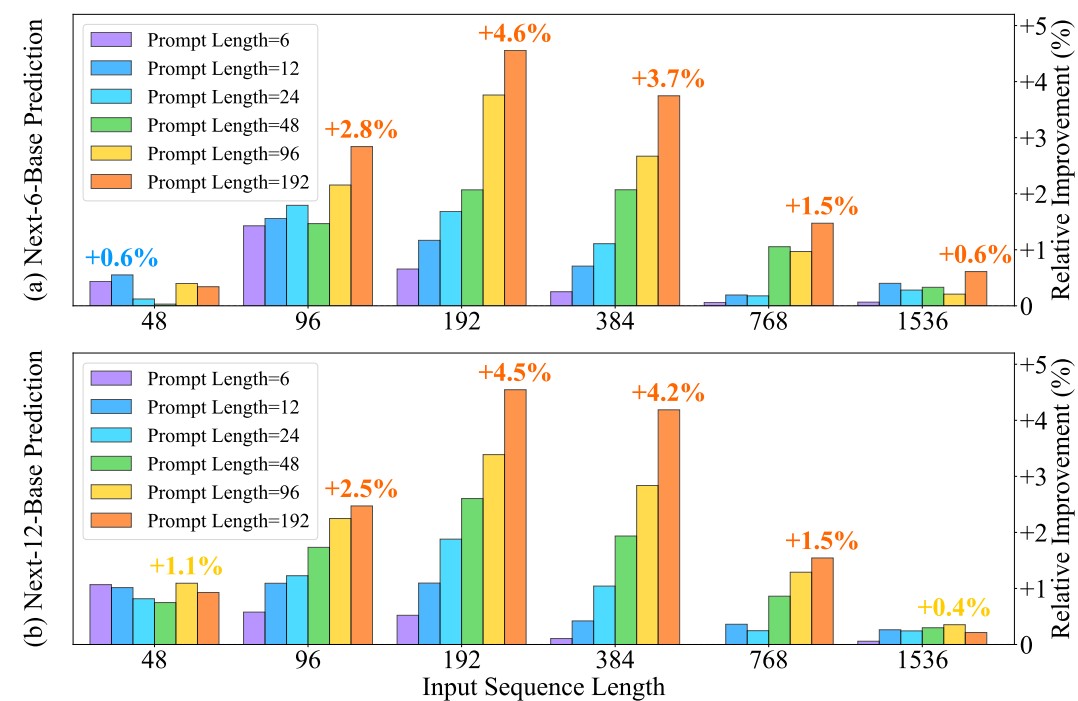

Figure 15: BasePrompt consistently enhances next-base prediction performance across varying input sequence lengths. As the input length increases, the relative accuracy gain over the baseline (original input) also grows. Panels (a) / (b) show the relative improvement results using the GENERator 3B model for predicting the next 6 / 12 bases given the input sequence, respectively.

Table 12: BasePrompt enhances generation quality in the next-6-base prediction task with a 192-base input. Compared to direct prediction on raw input, it yields nearly consistent gains across models even when generating a long 5′ prompt of 192 bases. Our method yields a relative accuracy improvement of 1.3–2.4% across all models. It also achieves relative reductions of 3.1–23.3% in GC Difference and 0.5–4.8% in 2-mer JSD for most models. Results are averaged across three subsets (Wu et al., 2025) to mitigate species imbalance.

| Model | Accuracy ↑ | | GC Difference ↓ | | JSD (2-mer) ↓ | |
|---|---|---|---|---|---|---|
| | Baseline | Ours | Baseline | Ours | Baseline | Ours |
| GENERator 1.2B | 37.7 | 38.5 | 0.0466 | 0.0474 | 0.1229 | 0.1204 |
| GENERator 3B | 40.0 | 40.9 | 0.0360 | 0.0349 | 0.1097 | 0.1044 |
| EVO2 1B base | 52.4 | 53.0 | 0.0174 | 0.0134 | 0.0872 | 0.0861 |
| EVO2 7B | 58.0 | 59.3 | 0.0195 | 0.0164 | 0.0709 | 0.0697 |
| EVO2 40B | 68.1 | 69.1 | 0.0178 | 0.0166 | 0.0504 | 0.0502 |

accuracy. However, BasePrompt can still steadily work on such variable sampling settings, and effectively facilitate the generation task.

The Python implementation of baseline prompting modes is shown as follows:

```python
def generate_prompts(mode:str, sequences: List[str], new_length: int):
    if mode == "RandomSeq":
        prompts = [
            "".join(random.choice(seq) for _ in range(new_length))
            for seq in sequences
        ]
    elif mode == "RandomBase":
```

```
 8        prompts = [
 9            "".join(random.choice('TAGC') for _ in range(new_length))
10            for _ in range(len(sequences))
11        ]
12    elif mode == "Repeat":
13        prompts = []
14        for seq in sequences:
15            temp_prompt = ""
16            while len(temp_prompt) < new_length:
17                temp_prompt = temp_prompt + seq[-new_length:]
18            prompts.append(temp_prompt[-new_length:])
19    elif mode in ["A", "T", "C", "G"]:
20        prompts = [
21            mode * new_length for _ in range(len(sequences))
22        ]
23    else:
24        raise ValueError(f"Invalid mode: {mode}")
25    return prompts
```

