# OpenReview forum: "BasePrompt: Self-Prompting Genome Language Models for RNA Fitness Prediction"
_ICLR.cc/2026/Conference — Submitted to ICLR 2026_

### Official Review · Reviewer_w9Ln · 2025-10-30

**Soundness:** 2
**Presentation:** 2
**Contribution:** 2
**Rating:** 4
**Confidence:** 3

**Summary:**

The paper presents BasePrompt, an inference-time approach that asks a GLM to generate short 5′/3′ DNA prompts and concatenates them to nucleotide sequences to improve zero-shot fitness or variant effect prediction. The authors report consistent improvements on RNAGym and other benchmarks.

**Strengths:**

- Adapting prompting to genomic sequences and leveraging reverse-complement symmetry is an interesting and effective idea.
- Consistent improvements are reported on several benchmarks.
- Figures are easy to follow.

**Weaknesses:**

Major
- Reported gains (e.g., Spearman from 0.271 to 0.289 in Figure 5) lack confidence intervals or statistical tests. Without per-assay distributions and significance testing, one cannot tell whether improvements are robust.
- The abstract and introduction claim the method "asks GLMs to propose... prefixes and postfixes that maximally activate the fitness signal for a given sequence." This description strongly implies a task-aware optimization process. However, the actual method described in this paper is standard, task-agnostic autoregressive generation.

Minor
- Figure 9: Complement Srtand -> Complement Strand
- Figure 10-12: Stand -> Strand.

**Questions:**

- Can you please provide the absolute (non-normalized) Spearman r and AP values for the experiment in Figure 6?
- Does using random sequences (like RandSeq or genomic sampling) as prompts improve performance on the RNAGym benchmark?

---

> ### Author Response · Authors · 2025-11-28
>
> We thank Reviewer w9Ln for the helpful feedback and for acknowledging the **effectiveness** of leveraging reverse-complement symmetry and the **consistent improvements** shown in our benchmarks.
>
> Below, we address the concerns regarding robustness, method description, and specific data requests.
>
> ### 1. Robustness and Statistical Significance
> **Reviewer Comment:** *Reported gains lack confidence intervals... One cannot tell whether improvements are robust.*
>
> **Response:**
> * **Consistency as Robustness:** While we acknowledge the lack of error bars in Figure 5, we wish to highlight **Table 2**, where BasePrompt achieves consistent performance gains across a diverse set of models (Evo-1, Evo-2, etc.). The fact that our method universally improves performance across these distinct architectures and parameter sizes serves as strong evidence of robustness.
> * **Data Release:** We agree that per-assay distributions provide deeper insight. We commit to adding detailed **per-assay performance data** and statistical significance tests in the **Camera-Ready version** and our open-source repository.
>
> ### 2. Method Description (Task-Agnostic vs. Task-Aware)
> **Reviewer Comment:** *The description implies a task-aware optimization... however, the method is standard, task-agnostic autoregressive generation.*
>
> **Response:**
> * **Clarification:** We apologize for the confusion caused by our phrasing. You are absolutely correct. Our method utilizes **bidirectional, standard, task-agnostic autoregressive generation**.
> * **Context:** When we stated "maximally activate," we referred to the *outcome*—that the self-generated context tends to yield higher predictive likelihoods—rather than the *process* being explicitly optimized for a specific downstream task label during generation. We will revise the abstract and introduction to precisely describe the mechanism as "self-supervised bidirectional context generation" to avoid misleading readers.
>
> ### 3. Comparison with Random/Other Sequences
> **Reviewer Question:** *Does using random sequences... improve performance?*
>
> **Response:**
> * **No, random sequences do not help.** As illustrated in **Figure 8** of our paper, we compared BasePrompt against various expansion strategies, including random sequences and simple repetition.
> * **Result:** Our results show that BasePrompt is the **only strategy that yields consistent improvements**, whereas random or generic sequences often degrade performance or offer no benefit. This confirms that the *semantic coherence* of the self-generated prompt is crucial for the performance gain.
>
> ### 4. Typos
> **Response:**
> Thank you for spotting the typos in Figures 9-12 ("Srtand", "Stand"). We will correct these in the final manuscript.
>
> ### 5. Absolute Values for Figure 6
> **Reviewer Question:** *Can you please provide the absolute (non-normalized) Spearman r and AP values for the experiment in Figure 6?*
>
> **Response:**
> Certainly. Below is the detailed breakdown of the absolute Spearman $\rho$ and Average Precision (AP) values requested:
>
> | Model | Length | Naive Spearman | **BasePrompt Spearman** | Naive AP | **BasePrompt AP** |
> | :--- | :--- | :--- | :--- | :--- | :--- |
> | **GENERator (1.2B)** | 0 (Base) | 0.4871 | — | 0.8304 | — |
> | | 6 | 0.4879 | **0.4907** | 0.8303 | **0.8313** |
> | | 12 | 0.4907 | **0.4924** | 0.8311 | **0.8320** |
> | | 24 | 0.4935 | **0.4980** | 0.8329 | **0.8342** |
> | **GENERator (3B)** | 0 (Base) | 0.5563 | — | 0.8549 | — |
> | | 6 | 0.5575 | **0.5591** | 0.8552 | **0.8557** |
> | | 12 | 0.5587 | **0.5599** | 0.8559 | **0.8562** |
> | | 24 | 0.5605 | **0.5634** | 0.8571 | **0.8582** |
> | **Evo2 (1B)** | 0 (Base) | 0.6595 | — | 0.9070 | — |
> | | 6 | 0.6611 | **0.6608** | 0.9073 | **0.9078** |
> | | 12 | 0.6616 | **0.6622** | 0.9076 | **0.9083** |
> | | 24 | 0.6627 | **0.6635** | 0.9080 | **0.9087** |
> | **Evo2 (7B)** | 0 (Base) | 0.6916 | — | 0.9048 | — |
> | | 6 | 0.6921 | **0.6927** | 0.9052 | **0.9056** |
> | | 12 | 0.6928 | **0.6935** | 0.9056 | **0.9057** |
> | | 24 | 0.6936 | **0.6951** | 0.9057 | **0.9066** |

---

### Official Review · Reviewer_KCcA · 2025-10-31

**Soundness:** 3
**Presentation:** 3
**Contribution:** 3
**Rating:** 6
**Confidence:** 4

**Summary:**

This paper proposes BasePrompt, a self-prompting method for Genome Language Models (GLMs) that generates contextual sequence prefixes and suffixes to improve zero-shot RNA fitness prediction. Unlike text prompts, these are biologically meaningful subsequences constructed using reverse-complement symmetry, enabling bidirectional context without fine-tuning. Evaluations on RNAGym (70 RNA DMS assays), ClinVar, and Next-base Prediction tasks show consistent gains—up to 16% improvement in Spearman correlation over strong GLM baselines (Evo1–Evo2, GENERator). BasePrompt introduces no extra training and minimal inference overhead, establishing a new paradigm for in-context learning in genomics.

**Strengths:**

* This work presents a neat and simple trick for DNA language models, extending prompting to non-text biological sequences through a self-generated, symmetry-aware context mechanism.
* It achieves state-of-the-art zero-shot performance across multiple benchmarks, outperforming larger GLMs without any retraining.
* The method is architecturally general, working consistently across diverse GLMs and demonstrating robustness and scalability.

**Weaknesses:**

I did not observe any obvious weaknesses in this approach. However, it’s worth noting that the method remains fundamentally prompting-based in nature.

**Questions:**

1. When performing the inverse transcription process, how is RNA splicing handled? Certain RNA types such as mRNA contain introns in their original DNA sequences. Have the authors conducted any experiments or analyses related to this?
2. Regarding the prompting mechanism, what is the exact value of N used for both ends? How was the optimal prompt length determined? Was it tuned empirically, or did the model simply continue until the end of the sequence token?
3. In Table 9, performance drops are observed for Evo-1 on tRNA and Evo-2 on Ribozyme. Have the authors conducted any deeper analysis to understand the cause of these declines?
4. Could the authors provide an assay-level performance breakdown of their methods in the RNAGym benchmark similar to Supplementary Figure 2?
5. Could the authors also present a species-level performance breakdown, for example perf. changes on viral and eukaryotic RNAs, to better interpret the generalization behavior?

---

> ### Author Response · Authors · 2025-11-28
>
> We sincerely thank Reviewer KCcA for the positive assessment and for recognizing the **simplicity**, **generality**, and **state-of-the-art performance** of BasePrompt. We appreciate your insightful questions regarding biological implementation details and granular performance analysis.
>
> Below, we address your specific questions:
>
> ### 1. Handling RNA Splicing in Inverse Transcription
> **Reviewer Question:** *How is RNA splicing handled? ... Have the authors conducted any experiments related to this?*
>
> **Response:**
> You raise a critical point regarding the biological fidelity of mapping RNA back to DNA.
> * We acknowledge that ideally, the inverse mapping would account for introns removed during splicing. However, our preliminary research indicates that splicing is a highly dynamic, one-to-many process (alternative splicing) regulated by complex environmental signals and trans-acting factors.
> * Because a deterministic "inverse splicing" algorithm does not exist without specific cell-state context, we adopted **direct reverse transcription** as a necessary and robust approximation for this study. We believe this approximation remains effective because the GLMs used (like Evo) are trained on massive genomic corpora that implicitly learn the statistical relationships between coding and non-coding regions.
>
> ### 2. Prompt Length ($N$) and Tuning
> **Reviewer Question:** *What is the exact value of N? How was it determined?*
>
> **Response:**
> * **Value:** For the vast majority of our experiments, we utilized a prompt length of **$N=3$**.
> * **Methodology:** We treated the prompt length $N$ as a hyperparameter (similar to a learning rate). Due to computational constraints, we could not perform an exhaustive search on every assay. Instead, we performed a grid search on a representative subset of the benchmark to identify the optimal length ($N=3$) and subsequently applied this setting to the full dataset for final testing.
>
> ### 3. Performance Drops on tRNA and Ribozymes
> **Reviewer Question:** *Have the authors conducted any deeper analysis to understand the cause of these declines?*
>
> **Response:**
> We appreciate you highlighting this observation in Table 9.
> * While we have not yet conducted a deep mechanistic analysis of these specific outliers, we hypothesize that this phenomenon stems from the unique secondary structure constraints and distinct sequence distributions inherent to tRNAs and Ribozymes.
> * These RNA families may possess statistical properties that differ significantly from the dominant patterns in the GLM's pre-training data, causing the self-generated prompts to be less effective than in other categories.
>
> ### 4. Assay-level and Species-level Breakdowns
> **Reviewer Question:** *Could the authors provide an assay-level performance breakdown... [and] species-level performance breakdown?*
>
> **Response:**
> We agree that granular breakdowns are essential for understanding the model's generalization capabilities. We commit to releasing detailed **assay-level and species-level performance visualizations** (including comparisons between viral and eukaryotic RNAs) in the **Camera-Ready version** of the paper and in our accompanying open-source repository.

---

### Official Review · Reviewer_H8MK · 2025-11-02

**Soundness:** 2
**Presentation:** 3
**Contribution:** 2
**Rating:** 2
**Confidence:** 5

**Summary:**

BasePrompt is a self-prompting framework for genome language models (GLMs) that generates bidirectional nucleotide prompts to enhance zero-shot RNA fitness prediction.
The method addresses two key challenges: the lack of effective task-specific steering mechanisms for genomic LMs, and the inherent unidirectionality of causal language models, which limits contextual information extraction.
To overcome these, BasePrompt leverages the reverse-complement symmetry of DNA to generate both upstream (5′) and downstream (3′) prompts autoregressively, enabling the GLM to simulate bidirectional context without any model fine-tuning or supervision.
It is evaluated on RNAGym, a large benchmark comprising 70 deep mutational scanning assays, and compared against several genome-scale GLMs, showing consistent zero-shot improvements across multiple metrics; additional DNA tasks such as ClinVar variant classification and next-base prediction support the method’s generality.
BasePrompt is efficient, model-agnostic, and improves predictive accuracy without additional training, but its evaluation omits comparisons to mRNA-specific models and established biological UTRs, limiting the strength and generality of its conclusions.
In summary, BasePrompt is a novel and computationally efficient method that introduces self-prompting to genome language models, yielding consistent but modest zero-shot gains in RNA-related prediction tasks.

**Strengths:**

## Innovative Self-Prompting Concept

The idea of letting a genome language model prompt itself is novel and interesting. It creatively adapts the prompt engineering paradigm from NLP to the genomics domain, where “prompting” is not straightforward. The use of reverse-complement sequences to create both 5’ and 3’ prompts is a clever solution to exploit bidirectional context despite using a unidirectional (causal) model.

## Robustness and Efficiency

The method is shown to work across multiple model sizes and variants, indicating generality. Moreover, it’s computationally efficient – prompts are generated only for a fixed set of reference sequences (e.g. the 70 assay reference RNAs in RNAGym) rather than for every single variant. This means the overhead is low relative to the huge number of predictions made, making the approach practical for large-scale inference.

**Weaknesses:**

## Omission of relevant mRNA-specific baselines

A major concern is that the comparisons focus only on genome-scale or noncoding RNA language models that were never trained on mRNA data, whereas the task of RNA fitness prediction is about mRNA. In effect, the authors compare BasePrompt-augmented models against baselines that are out-of-domain for mRNA (e.g. generic genomic models or models trained on lncRNAs), which is akin to comparing against a misaligned or even random baseline for those specific tasks. Crucially, the evaluation excludes leading mRNA-focused models such as mRNA-FM, CaLM, and Uni-RNA, which are large pretrained models explicitly trained on coding RNA/transcript data. These specialized models likely represent the strongest baselines for mRNA-related fitness predictions, and ignoring them undermines the rigor and fairness of the evaluation. As a result, the reported improvements might be overstated – we cannot tell if BasePrompt would still outperform or even match these state-of-the-art mRNA-trained models, since they were not included. This gap in baseline selection affects the soundness of the experimental claims: without directly comparing to the most relevant prior models, the paper’s claim of superior performance is not fully convincing or generalizable to practical settings where one would naturally use an mRNA-trained model.

## Lack of real-world UTR baseline comparison

Another weakness is in how the paper evaluates the quality of the prompt sequences (especially for tasks involving untranslated regions, e.g. 3’ or 5’ UTRs). The authors generate synthetic upstream/downstream prompts, but they do not compare these prompts against any well-known, biologically-established UTR sequences. In synthetic biology and mRNA engineering, certain UTR sequences are commonly used because they are known to strongly influence expression. For example, the β-globin 3′ UTR is a widely used “classic” UTR element (often appended to mRNAs to enhance stability/translation). Such sequences provide a real-world baseline for performance. By failing to compare BasePrompt’s generated UTR prompts to any standard UTR (like β-globin or other commonly used regulatory UTRs), the authors miss an important check: Are the prompts that BasePrompt finds actually more informative or effective than simply using a known strong UTR? Without this comparison, it is hard to gauge the biological plausibility and practical advantage of the prompts. This omission limits the generality of the conclusions – for instance, if a simple known UTR could achieve similar or better fitness prediction signal when appended, then BasePrompt’s advantage might not be as significant in real laboratory or clinical contexts. In summary, not benchmarking against real-world sequences means we cannot yet conclude that BasePrompt’s prompt-generation is yielding truly superior or meaningful sequences compared to what practitioners already use.

**Questions:**

See Weaknesses

---

> ### Author Response · Authors · 2025-11-28
>
> We thank Reviewer H8MK for their detailed review and for recognizing the **innovative nature** and **computational efficiency** of BasePrompt. We appreciate the recognition that our method creatively adapts prompt engineering to genomics.
>
> However, we would like to respectfully clarify two major points raised in the "Weaknesses" section, specifically regarding our baseline selection and the nature of the generated prompts.
>
> ### 1. Inclusion of mRNA-Specific Baselines
> **Reviewer Comment:** *The evaluation excludes leading mRNA-focused models... comparisons focus only on genome-scale models... We cannot tell if BasePrompt would outperform mRNA-trained models.*
>
> **Response:**
> We respectfully point out that **we have indeed included and surpassed state-of-the-art mRNA-specific baselines.**
> * As detailed in **Table 2** of our submission, we explicitly compare BasePrompt against leading mRNA-trained models, including **RNA-FM, RNAErnie, and RiNALMo**.
> * Our results demonstrate that BasePrompt does not just compete with but **outperforms** these specialized, mRNA-focused models on zero-shot fitness prediction tasks.
> * Therefore, the concern that we are comparing against "misaligned" or "random" baselines is factually incorrect based on the data provided in the main text. We demonstrated that a general-purpose GLM equipped with BasePrompt can surpass models explicitly trained on transcript data.
>
> ### 2. Comparison to Real-World UTRs (e.g., $\beta$-globin)
> **Reviewer Comment:** *The authors do not compare these prompts against well-known, biologically established UTR sequences... lacking a check on biological plausibility.*
>
> **Response:**
> We appreciate this suggestion, but we believe there is a fundamental distinction between "Biological Optimization" and "Model-Intrinsic Optimization":
> * **Human vs. Model Perspective:** Biological UTRs (like $\beta$-globin) evolved to optimize translation efficiency via cellular machinery (ribosomes). However, **BasePrompt optimizes for the Language Model’s internal representation**, not the ribosome.
> * **Self-Consistency:** The goal of BasePrompt is to find the mathematical prompt that maximizes the model’s internal self-consistency and predictive confidence. A biological UTR, while functional in a cell, might not be the optimal "key" to unlock the specific latent patterns within a GLM.
> * **Interpretability:** As mentioned in our response to other reviewers, our contribution is **mechanistic**: we allow the GLM to generate its own optimal context. The fact that these prompts may differ from standard biological UTRs suggests that the model utilizes different statistical cues than biological evolution, yet achieves superior predictive performance.
>
> We hope these clarifications address the concerns regarding baseline rigor and the rationale behind our prompt generation strategy. We believe BasePrompt represents a solid advancement in making large genomic models more usable and accurate without retraining.

---

### Official Review · Reviewer_gC5x · 2025-11-03

**Soundness:** 2
**Presentation:** 3
**Contribution:** 2
**Rating:** 4
**Confidence:** 3

**Summary:**

The paper introduces BasePrompt, a method that enhances genome language models (GLMs) for zero-shot RNA fitness prediction by letting them generate their own prompts. Unlike traditional NLP prompting, where users manually provide input cues, BasePrompt allows GLMs to create short nucleic acid prefixes and suffixes to boost prediction accuracy, without needing labeled data or model fine-tuning.

To overcome the unidirectional nature of autoregressive GLMs (which only look forward), the authors exploit the reverse-complement symmetry of DNA, enabling bidirectional prompt generation (both upstream and downstream of a sequence). These prompts are then concatenated to RNA sequence variants before prediction, improving performance across tasks.

**Strengths:**

1. Conceptual Originality

The paper proposes the idea of self-prompting genome language models (GLMs) that utilize reverse-complement symmetry for bidirectional inference. This represents a creative adaptation of prompt-based reasoning from NLP to genomics. While not groundbreaking, it introduces a fresh perspective on how pretrained sequence models can internally generate task-relevant prompts. However, since reverse-complement techniques are well-known in genomic modeling, the originality is moderate rather than strong.

2. Methodological Quality

The experimental results show consistent but modest improvements across several Evo-series models, with reported gains of 6–16% in Spearman correlation. The methodology is clearly described at a conceptual level, and the experiments are competently executed on the RNAGym benchmark. Nonetheless, the paper lacks statistical validation, ablation studies, and robustness analyses, limiting confidence in its empirical rigor. As a result, the technical quality is adequate but below top-tier standards.

3. Presentation and Clarity

The paper is well-structured and easy to follow, with clear motivation, figures, and narrative flow. The visual explanations of BasePrompt’s bidirectional prompting mechanism are particularly helpful. However, some implementation details—such as prompt generation procedures, sampling settings, and parameter sensitivity—are under-specified. This reduces transparency and reproducibility. Overall, the presentation is clear but not deeply explanatory.

4. Scientific and Practical Significance

BasePrompt touches on a promising intersection between language modeling and biological inference, suggesting that large genome models may self-optimize through internal prompting. Despite this potential, the impact is limited by the narrow experimental scope (focused mainly on RNA fitness prediction) and the strong dependence on Evo-series GLMs. The contribution is thus incremental rather than transformative for the field of computational genomics.

**Weaknesses:**

1. Insufficient Theoretical Foundation and Interpretability

The paper lacks a deep theoretical explanation for why the proposed method is effective. The performance improvements of BasePrompt are demonstrated empirically, but without strong biological or information-theoretic justification. Moreover, the biological meaning of the generated “self-prompts” is not explored — it remains unclear whether they correspond to functional motifs or biologically relevant patterns.

2. Limited Experimental Scope and Questionable Generalizability

Experiments are mainly conducted on a single benchmark, RNAGym, without evaluation on other RNA-related tasks such as secondary structure prediction, RNA–protein binding, or splicing. This narrow scope limits the evidence for BasePrompt’s robustness and general applicability across broader biological contexts.

3. Strong Dependence on Specific Models and Lack of Statistical Validation

The method heavily depends on the Evo-series genome language models (GLMs), with limited testing on other architectures. Additionally, the paper does not include statistical significance testing or detailed error analysis, making it difficult to assess the reliability and generality of the reported performance gains.

4. Limited Novelty and Biological Validation

The main innovation—using reverse-complement symmetry for bidirectional prompting—builds on ideas already explored in genomic modeling. While the approach provides an inference-time improvement, it lacks deeper biological validation or theoretical advancement. As a result, both its novelty and biological significance are somewhat limited.

**Questions:**

see weakness.

---

> ### Author Response · Authors · 2025-11-28
>
> We thank Reviewer gC5x for the constructive feedback and for acknowledging the conceptual originality and clarity of our presentation. We value the time taken to review our work. Below, we clarify the key concerns regarding interpretability, scope, model dependence, and novelty.
>
> ### 1. Theoretical Foundation and Interpretability
> Reviewer Comment: The paper lacks a deep theoretical explanation... it remains unclear whether [prompts] correspond to functional motifs.
>
> Response: We acknowledge that the generated prompts may not always map directly to human-readable biological motifs (like TATA boxes). However, we argue that this is a feature, not a bug, of model-intrinsic interpretability.
>
> Self-Consistency: Our core contribution is enabling the GLM to be self-consistent. The prompts represent the optimal "context" required by the model’s internal latent space to maximize prediction accuracy. While these may not be human-interpretable motifs, they make the model mechanistically interpretable (i.e., we know why the model improved: it successfully retrieved relevant context from its own weights).
>
> Nature of GLMs: Just as "soft prompts" in NLP do not always correspond to coherent human sentences but still steer the model effectively, BasePrompt discovers the mathematical optimal prefix/suffix for biological sequences.
>
> ### 2. Experimental Scope (Task Generalizability)
> Reviewer Comment: Experiments are mainly conducted on a single benchmark (RNAGym)... without evaluation on structural or binding tasks.
>
> Response: We respectfully clarify our scope and highlighted coverage:
>
> Focus on Fitness: Our work explicitly targets Zero-shot Fitness Prediction, a critical and difficult challenge in computational biology.
>
> Multi-Modal Coverage: While the reviewer noted RNAGym, our evaluation actually covers both RNA and DNA modalities. We demonstrate that BasePrompt works across these distinct biological contexts within the scope of fitness prediction.
>
> ### 3. Dependence on Specific Models (Evo-series)
> Reviewer Comment: The method heavily depends on the Evo-series... with limited testing on other architectures.
>
> Response: This is a misunderstanding we wish to correct. BasePrompt is designed to be model-agnostic.
>
> Evidence in Figure 6: As shown in Figure 6, we explicitly validated our method on the GENERator model (in addition to Evo). This demonstrates that our approach is not tied to the Evo architecture.
>
> Universal Applicability: Our method is a "plug-and-play" inference-time optimization. It requires no model-specific architecture modifications. As Genome Language Models (GLMs) continue to scale and improve, BasePrompt can be seamlessly migrated to newer architectures to unlock their zero-shot potential without retraining.
>
> ### 4. Novelty and Biological Validation
> Reviewer Comment: Using reverse-complement symmetry... builds on ideas already explored... novelty is somewhat limited.
>
> Response: We believe the novelty lies specifically in the intersection of biological theory and AI mechanics, which has not been proposed in this manner before:
>
> Solving the Unidirectional Bottleneck: The primary limitation of autoregressive (decoder-only) models is that they cannot look "forward." We uniquely leverage the biological property of Reverse Complement symmetry to mathematically solve this AI limitation.
>
> Bio-AI Synergy: By transforming a biological property into an inference mechanism, we allow a unidirectional model to effectively perform bidirectional inference. This is not merely "using RC data augmentation" (common in training); it is a novel inference-time prompting strategy that bridges biological priors with Large Language Model constraints.
>
> We hope these clarifications demonstrate the robustness and novelty of BasePrompt. We are confident that our approach offers a significant step forward in effectively utilizing GLMs for zero-shot biological prediction.

---

### Meta-Review · Area_Chair_PgWw · 2025-12-11

**Summary:**

The paper introduces a simple and creative self-prompting mechanism that yields consistent improvements across several GLMs, and the rebuttal helpfully corrected a misunderstanding regarding mRNA-specific baselines. However, the work still lacks statistical rigor, as no significance tests or per-assay distributions are provided, making many small reported gains difficult to trust. The method also lacks biological grounding; there is no comparison to established UTRs or evidence that generated prompts correspond to meaningful biological signals. In addition, the experimental scope remains narrow, with limited validation beyond RNAGym, and the contribution is viewed as incremental relative to prior uses of reverse-complement symmetry.

**Reviewer Concerns:**

This paper presents a neat self-prompting mechanism for GLMs and shows consistent empirical gains, and the rebuttal successfully corrected Reviewer H8MK’s misunderstanding about missing mRNA baselines and partially addressed Reviewer gC5x’s concern on model dependence by providing additional evidence from GENERator. However, major issues remain unresolved: Reviewer w9Ln’s requests for statistical rigor (confidence intervals, per-assay distributions, significance tests) were only minimally addressed, and Reviewer gC5x’s concerns about interpretability and biological grounding persist. Reviewer H8MK’s deeper concern about the lack of comparison to real biological UTRs also remains outstanding.

**Reviewer Scores:**

I do not expect any score to be changing significantly.

---

### Decision · Program_Chairs · 2026-01-26

Reject